# Learn to Vaccinate: Combining Structure Learning and Effective Vaccination for Epidemic and Outbreak Control

Sepehr Elahi [* 1]   Paula Mürmann [* 1]   Patrick Thiran [1]

## Abstract

The Susceptible-Infected-Susceptible (SIS) model is a widely used model for the spread of information and infectious diseases, particularly non-immunizing ones, on a graph. Given a highly contagious disease, a natural question is how to best vaccinate individuals to minimize the disease's extinction time. While previous works showed that the problem of optimal vaccination is closely linked to the NP-hard *Spectral Radius Minimization* (SRM) problem, they assumed that the graph is known, which is often not the case in practice. In this work, we consider the problem of minimizing the extinction time of an outbreak modeled by an SIS model where the graph on which the disease spreads is unknown and only the infection states of the vertices are observed. To this end, we split the problem into two: learning the graph and determining effective vaccination strategies. We propose a novel inclusion-exclusion-based learning algorithm and, unlike previous approaches, establish its sample complexity for graph recovery. We then detail an optimal algorithm for the SRM problem and prove that its running time is polynomial in the number of vertices for graphs with bounded treewidth. This is complemented by an efficient and effective polynomial-time greedy heuristic for any graph. Finally, we present experiments on synthetic and real-world data that numerically validate our learning and vaccination algorithms.

## 1. Introduction

Spreading models are essential frameworks for understanding how information, diseases, or behaviors propagate through networks (Nowzari et al., 2016). These models capture the complex interactions between agents, helping to predict dissemination patterns and devise strategies to control unwanted spread (Lokhov & Saad, 2017). Among these models, the *Susceptible-Infected-Susceptible* (SIS) framework is particularly well-suited for representing scenarios where agents can undergo multiple cycles of infection and recovery. While simple in structure, the interactions between neighboring agents in an SIS model can give rise to complex behavior, which makes predicting and controlling its dynamics challenging.

The SIS model is represented using a contact graph where each vertex corresponds to an agent that can be in one of two states: *susceptible* or *infected*. The state of each vertex evolves probabilistically, influenced by the infection statuses of its neighboring vertices, allowing for multiple cycles of infection and recovery. In recent years, SIS models have seen applications in fields such as the modeling of credit and financial markets (Chen & Fan, 2023; Barja et al., 2019), rumor spreading in social networks (Dong & Huang, 2018), malware attacks on computer networks (Märtens et al., 2016), and epidemiology (Nowzari et al., 2016; Grandits et al., 2019).

To mitigate the adverse effects of contagions modeled by the SIS framework, it is often necessary to perform strategic interventions. These interventions, in the form of regulatory interventions (Chen & Fan, 2023), installation of updates and patches (Muthukumar et al., 2024), vaccinations or quarantines of vertices (i.e., person or community), are designed to reduce transmission rates, limit the reach of the contagion, and accelerate its extinction. However, devising effective intervention strategies is particularly challenging when the underlying graph is unknown, a common scenario in practice (Rosenblatt et al., 2020).

We provide a motivating example in the field of epidemiology. While we adopt the epidemiological terminology in this work, our approaches are general and can be readily applied to any other domain that utilizes the SIS framework.

**Motivating Example.** Consider the spread of cholera (*Vibrio cholerae*), a highly contagious waterborne disease that individuals can contract multiple times, making it well-suited

---

*Equal contribution  [1]EPFL, Switzerland. Correspondence to: Sepehr Elahi <sepehr.elahi@epfl.ch>, Paula Mürmann <paula.murmann@epfl.ch>.

*Proceedings of the 42nd International Conference on Machine Learning*, Vancouver, Canada. PMLR 267, 2025. Copyright 2025 by the author(s).

to the SIS model (Ryan & Calderwood, 2000). In many regions, the exact network of human interactions and environmental factors that facilitate cholera transmission is often not fully mapped, compelling public health officials to rely on observed infection data to infer these interaction patterns. By learning this underlying contact network, public health authorities can implement ring vaccination strategies, which involve targeting vaccinations to individuals to effectively reduce transmission rates and extinction time (Ali et al., 2016).

In this paper, we focus on determining effective vaccination strategies for an SIS epidemic when the underlying graph is unknown and only infection data is observed. Inspired by real-world approaches like ring vaccination, our approach combines network learning with a targeted vaccination strategy to minimize disease extinction time and control contagion spread.

**Structure Learning.** Most research concerning epidemic structure recovery has been focused on *Susceptible-Infected-Removed* (SIR) epidemic models, where each vertex can be infected only once before becoming inert. In SIR models, multiple infection instances, or *cascades*, are necessary to accurately infer the underlying network structure (Netrapalli & Sanghavi, 2012), with a common approach being maximum likelihood estimation (MLE) (Gomez-Rodriguez et al., 2012; Myers & Leskovec, 2010; Gray et al., 2020).

In contrast, learning the network structure of an SIS model does not require multiple cascades since vertices can experience repeated infections. This allows for the observation of more correlations between neighbors over time. However, the absence of a directional infection flow in SIS models introduces additional noise, complicating the learning.

To the best of our knowledge, the only work on structure learning in SIS models is an MLE-based approach by Barbillon et al. (2020). Their approach can recover both likely edges and infection parameters, but comes without theoretical guarantees and assumes a vertex is infected by only one neighbor at a time, which is a strong assumption in practice.

We do not make this restrictive assumption in our work, and instead propose a novel learning algorithm inspired by Ising model learning from Bresler (2015) that leverages the fact that stronger correlations occur between infection states of neighboring vertices than those of non-neighbors.

**Vaccination Strategies.** To suppress an undesirable SIS process, effective vaccination strategies are essential. Unlike SIR models, SIS processes allow for repeated infections, making them more challenging to eradicate. The extinction time of an SIS process is closely related to the spectral radius of the underlying graph (Ahn & Hassibi, 2014; Ruhi et al., 2016), leading to approaches that employ spectral radius

minimization as a vaccination heuristic (Van Mieghem et al., 2011; Kiji et al., 2022). However, these methods assume the complete removal of either the vaccinated vertices or a collection of their incident edges. More importantly, no work has addressed the problem of vaccinations when the graph structure is unknown.

Furthermore, existing work often considers continuous-time SIS models, where vaccinations are applied continuously and can boost vertices' recovery rates (Abad Torres et al., 2017; Scaman et al., 2016; Drakopoulos et al., 2014). In contrast, we adopt a discrete-time SIS model with simultaneous updates and focus on one-time vaccinations that only reduce infection probabilities. This model allows for both sparser observations of the states and fewer interventions, which is often more applicable in practice.

**Original Contributions.** We address the challenge of devising effective vaccination strategies in an ongoing SIS process where the underlying graph is unknown. We approach this problem by decomposing it into two tasks: first learning the network from observed infection states, and then designing effective vaccination strategies based on the inferred graph. Our primary contributions are as follows:

- We formally introduce the Vaccinating an Unknown Graph (VUG) problem, which aims to minimize the extinction time of an SIS process on an unknown graph through strategic vaccinations (Section 2).

- We propose a novel inclusion-exclusion algorithm for learning the graph of an SIS epidemic, without making the restrictive assumption of the previous approach, and with learning guarantees (Section 3).

- Leveraging the Spectral Radius Minimization problem as a proxy for effective vaccination, we develop an exact dynamic programming-based vaccination strategy that runs in polynomial time for graphs with bounded treewidth. Additionally, we present a simple, yet effective, polynomial-time greedy heuristic (Section 4).

- We conduct comprehensive experiments on real and synthetic data to evaluate the performance of our learning and vaccination algorithms. Our results demonstrate significant improvements over existing baseline methods in terms of vaccination efficacy (Section 5).

## 2. Problem Formulation

We first introduce the SIS model, followed by a definition of vaccinations, and finally the problem of vaccinating vertices to minimize the extinction time of the SIS process.

**SIS Infection Model.** We consider a *Susceptible-Infected-Susceptible* (SIS) model of disease propagation, set on a

graph $\mathcal{G} = (V, E)$. We also use $V(\mathcal{G})$ to denote the set of vertices of $\mathcal{G}$ when specificity is needed. The graph $\mathcal{G}$ is composed of $n$ vertices $V = [n] \coloneqq \{1, \ldots, n\}$ representing individuals or entities, and a set of undirected edges $E$ that capture the pathways through which the disease can spread. Given the edge $(i, j) \in E$, we refer to vertices $i$ and $j$ as *neighbors*, and we use $\mathcal{N}(j) \coloneqq \{i \mid (i, j) \in E\}$ to denote the neighborhood of vertex $j$. We use $\Delta$ to denote the maximum degree of the graph, i.e., $\Delta = \max_{i \in V} |\mathcal{N}(i)|$. Finally, we denote by $\rho(\mathcal{G})$ the *spectral radius* (i.e., largest eigenvalue) of the adjacency matrix of the graph $\mathcal{G}$.

We consider a discrete-time SIS model that proceeds over rounds (time steps) indexed by $t \in [T] = \{1, \ldots, T\}$, where each vertex $i$ can be in one of two states: *susceptible* or *infected*. The state of vertex $i$ in round $t$ is denoted by $Y_i^{(t)} \in \{0, 1\}$ (1 for infected and 0 for susceptible). Furthermore, we use $Y_S^{(t)} \in \{0, 1\}^{|S|}$ to denote the state vector of a vertex set $S \subseteq V$. We denote by $Y^{(t)} = \left\{ Y_i^{(t)} \right\}_{i=1}^n$ the state vector of all vertices in round $t$, and $Z^{(t)} \in \{0\} \cup [n]$ to denote the number of infected vertices in round $t$, i.e., $Z^{(t)} = \sum_{i=1}^n Y_i^{(t)}$.

The SIS model is characterized by three key parameters: the seed probability $p_{\text{init}}$, the infection probability $p_{\text{inf}}$, and the recovery probability $p_{\text{rec}}$. In the initial round $t = 0$, each vertex is infected independently with probability $p_{\text{init}}$. As the infection progresses, an infected vertex transmits the disease to its susceptible neighbors with probability $p_{\text{inf}}$, while each infected vertex recovers and becomes susceptible again with probability $p_{\text{rec}}$.[1] Below is a table summarizing the conditional probabilities of the SIS model:

| Event | Description | Probability |
|---|---|---|
| $Y_i^{(t+1)} = 1 \mid Y_i^{(t)} = 1$ | $i$ remains infected | $1 - p_{\text{rec}}$ |
| $Y_i^{(t+1)} = 0 \mid Y_i^{(t)} = 1$ | $i$ becomes susceptible | $p_{\text{rec}}$ |
| $Y_i^{(t+1)} = 1 \mid Y_i^{(t)} = 0$ | $i$ becomes infected | $1 - \prod_{j \in \mathcal{N}(i)}(1 - p_{\text{inf}} \cdot Y_j^{(t)})$ |
| $Y_i^{(t+1)} = 0 \mid Y_i^{(t)} = 0$ | $i$ remains susceptible | $\prod_{j \in \mathcal{N}(i)}(1 - p_{\text{inf}} \cdot Y_j^{(t)})$ |

We say that an SIS infection is *extinct* at step $t$ if $Z^{(t)} = 0$. We denote the *extinction time*, if it exists, by $\tau \coloneqq \min\{t \in [T] \mid Z^{(t)} = 0\}$, and $\tau = \infty$, if otherwise.

**Vaccination.** In each round $t$, the vaccination strategy can vaccinate a subset of vertices $R_t \subseteq V$ to lower their infection probability. Formally, the strategy is given an overall budget $K \in \mathbb{Z}^+$ and can vaccinate at most $K$ vertices in all rounds, i.e., $\sum_{t=1}^T |R_t| \leq K$. If a vertex $i$ is vaccinated in round $t'$, its infection probability is reduced by a factor of $\alpha \in [0, 1]$ in all future rounds, i.e., if $i \in R_{t'}$, then $\mathbb{P}\left(Y_i^{(t+1)} = 1 \mid Y_i^{(t)} = 0\right) = 1 - \prod_{j \in \mathcal{N}(i)}(1 - \alpha p_{\text{inf}} \cdot Y_j^{(t)})$

---

[1]Note that neighbors infect each other independently and infected vertices heal independently of one another.

for all $t > t'$.

**VUG Problem.** The goal of the Vaccinating an Unknown Graph (VUG) problem is to select vertices to vaccinate in order to minimize the expected extinction time, while only observing the infection states $Y_t$ in each round $t$, i.e., the set of edges $E$ is unknown.

Formally, let $\pi \colon [T] \times \{0, 1\}^{nT} \to V$ denote a vaccination strategy, where $\pi(t, Y_t) = R_t$ specifies the vertices to be vaccinated in round $t$. The objective is then to minimize the expected extinction time, subject to a vaccination budget of $K$: $\min_\pi \mathbb{E}[\tau_\pi]$ such that $\sum_{t=1}^T |R_t| \leq K$, where $\tau_\pi$ represents the extinction time under strategy $\pi$. Note that the VUG problem is online: the agent observes the infection states over time and must decide when and whom to vaccinate, subject to the global budget $K$.

## 3. Learning the Graph

First, we present our approach for learning the graph from the observed infection states and defer the discussion of our vaccination strategies to the next section.

At a high-level, our learning approach makes use of the idea that the infection state of a vertex $j$ at time $t$ is more likely to be influenced by the infection states of its neighbors at time $t - 1$ than by the infection states of other vertices. We now formalize this idea.

### 3.1. Direct and Conditional Influence

Inspired by the work of Bresler (2015) on structure learning in Ising models, we introduce two measures of how vertex $i$ influences vertex $j$ in round $t$: the *direct influence* (DI), $\mu_{j|i}^{(t)}$, and the *conditional influence* (CI), $\nu_{j|i,y_S}^{(t)}$. Intuitively, DI captures the probability that $i$ directly infects $j$, while CI measures how much $i$'s infection state affects $j$'s infection probability while conditioning on a set $S \subseteq V \setminus \{i, j\}$. As we will show, if $S$ separates $i$ from $j$ in the graph, then $i$'s impact on $j$ vanishes.

Formally, we define

$$\mu_{j|i}^{(t)} \coloneqq \mathbb{P}\left(Y_j^{(t+1)} = 1 \,\Big|\, Y_j^{(t)} = 0, \, Y_i^{(t)} = 1\right),$$

and, for any $y_S \in \{0, 1\}^{|S|}$,

$$\nu_{j|i,y_S}^{(t)} \coloneqq$$
$$\mathbb{P}\left(Y_j^{(t+1)} = 1 \,\Big|\, Y_j^{(t)} = 0, \, Y_i^{(t)} = 1, \, Y_S^{(t)} = y_S\right)$$
$$- \mathbb{P}\left(Y_j^{(t+1)} = 1 \,\Big|\, Y_j^{(t)} = 0, \, Y_i^{(t)} = 0, \, Y_S^{(t)} = y_S\right).$$

Using these quantities, we devise an inclusion-exclusion-based algorithm to learn the neighbors of a vertex $j$. More

precisely, we first learn a superset of the neighbors of $j$, and then reject all non-neighbors. We explicitly explain both parts below, providing all missing proofs in Appendix D.3.

**Inclusion.** If $j$ is susceptible at time $t$ and its neighbor $i$ is infected, then $i$ infects $j$ with probability $p_{inf}$ in the next round. Then, as $j$ may have other infected neighbors, we get the following lower bound for $\mu_{j|i}^{(t)}$:

**Lemma 3.1.** *If $i, j \in V$ are neighbors, then*

$$\mu_{j|i}^{(t)} \geq p_{inf} \quad \forall t \in [T].$$

We use this bound to construct a superset of the neighbors of $j$ by including all vertices $i$ that satisfy $\mu_{j|i}^{(t)} \geq p_{inf}$.

**Exclusion.** To reject non-neighbors from a candidate neighbor set, observe that if $k$ is not a neighbor of $j$, then conditioning on any set $S \subseteq V \setminus \{j, k\}$ that separates $k$ from $j$ forces the conditional influence of $k$ on $j$ to drop to zero. Intuitively, once we fix $j$'s neighbors, $j$ is isolated from $k$. Hence, applying the local Markov property, $j$'s transition probabilities in the next round only depends on its neighboring states, but not on $k$.

We state and prove the above observation as a lemma below.

**Lemma 3.2.** *Let $j, k \in V$ be non-neighbors and let $S \subseteq V \setminus \{j, k\}$ be a superset of neighbors of $j$, i.e., $\mathcal{N}(j) \subseteq S$. Then*

$$\nu_{j|k,y_S}^{(t)} = 0 \quad \forall t \in [T], \; y_S \in \{0,1\}^{|S|}.$$

Finally, to ensure that using $\nu_{j|i,y_S}^{(t)}$ to exclude non-neighbors does not mistakenly remove a neighbor, we show that a neighbor's CI on vertex $j$ is bounded away from zero.

**Lemma 3.3.** *Let $i, j \in V$ be neighbors and let $S \subseteq V \setminus \{j\}$ be a superset of neighbors of $j$, i.e., $\mathcal{N}(j) \subseteq S$. Then, for any $y_{S \setminus \{i\}} \in \{0,1\}^{|S|-1}$, $t \in [T]$*

$$\nu_{j|i,y_{S \setminus \{i\}}}^{(t)} \geq p_{inf}(1 - p_{inf})^{\Delta - 1}.$$

We summarize the above lemmas in the following corollary.

**Corollary 3.1.** *Let $i, j \in V$ and $S \subseteq V \setminus \{j\}$ such that $\mathcal{N}(j) \subseteq S$, then for any $t \in [T]$ and any $y_{S \setminus \{i\}} \in \{0,1\}^{|S|-1}$:*

$$i \in \mathcal{N}(j) \Leftrightarrow \nu_{j|i,y_{S \setminus \{i\}}}^{(t)} \geq p_{inf}(1 - p_{inf})^{\Delta - 1}.$$

*Proof.* ($\Rightarrow$) By Lemma 3.3 and ($\Leftarrow$) by Lemma 3.2. $\square$

Next, we establish how we estimate the direct and conditional influence from observations.

### 3.2. Estimation of Direct and Conditional Influence

For the purposes of theoretical analysis, including learning guarantees and estimation of DI and CI, we follow the approach of Van De Bovenkamp & Van Mieghem (2014) and work with a modified ergodic Markov chain, $\{\Psi^{(t)}\}_{t \in [T]}$. This chain has the same transition dynamics as the original SIS process $\{Y^{(t)}\}$, except that upon reaching the all-zero (extinction) state, it immediately restarts from a predefined initial distribution (see Appendix D.1 for details).

Crucially, the original SIS process $\{Y^{(t)}\}$ and the modified chain $\{\Psi^{(t)}\}$ can be coupled up to the extinction time of $\{Y^{(t)}\}$. As a result, any analysis or estimation conducted on $\{\Psi^{(t)}\}$ before its first restart captures the behavior of the original process in the pre-extinction regime, which is precisely the regime relevant for the VUG problem.

**Stationary Assumption.** The direct and conditional influences introduced earlier are time-specific and thus cannot be reliably estimated from a single realization of either SIS process. However, empirical evidence suggests that the original process $\{Y^{(t)}\}$ quickly reaches a meta-stable state, which corresponds to the stationary distribution of the modified process $\{\Psi^{(t)}\}$ (Van De Bovenkamp & Van Mieghem, 2014).

**Assumption 3.1.** *The dataset $\mathcal{D} = \{\Psi^{(t)}\}_{t \in T_D}$ used for our estimations is drawn from the unique stationary distribution of the modified ergodic Markov chain $\{\Psi^{(t)}\}$.*

As shown in Figure 1 and further discussed in Appendix A.1, our simulations demonstrate that the SIS process $\{Y^{(t)}\}$ consistently enters a meta-stable regime within a few hundred rounds across a broad range of parameters. This observation, along with the following remark, justifies both the stationarity assumption and the use of the modified process $\{\Psi^{(t)}\}$ as a proxy for $\{Y^{(t)}\}$.

**Remark 3.1.** *Using the modified process $\{\Psi^{(t)}\}$ is natural in the context of the VUG problem: the SIS process $\{Y^{(t)}\}$ either converges to a meta-stable state, all the while coupled to process $\{\Psi^{(t)}\}$, or it goes extinct. The VUG problem is most challenging, and vaccinations critical, in the former scenario, where the infection persists and requires strategic vaccination for eradication.*

The use of $\{\Psi^{(t)}\}$ and Assumption 3.1 will be further justified in Section 5, where we do not artificially enforce ergodicity or stationarity, and yet our learning algorithm sufficiently recovers the graph.

While as a consequence of Assumption 3.1, DI and CI become stationary over time (i.e., for $t, t' \in T_D$, $\mu_{j|i}^{(t)} = \mu_{j|i}^{(t')}$ and $\nu_{j|i,y_S}^{(t)} = \nu_{j|i,y_S}^{(t')}$), they still depend on a one-step transition between the conditioned and observed states. To capture this, we will use $\Psi^{(\cdot)}$ and $\Psi^{(\cdot+1)}$ to denote a state

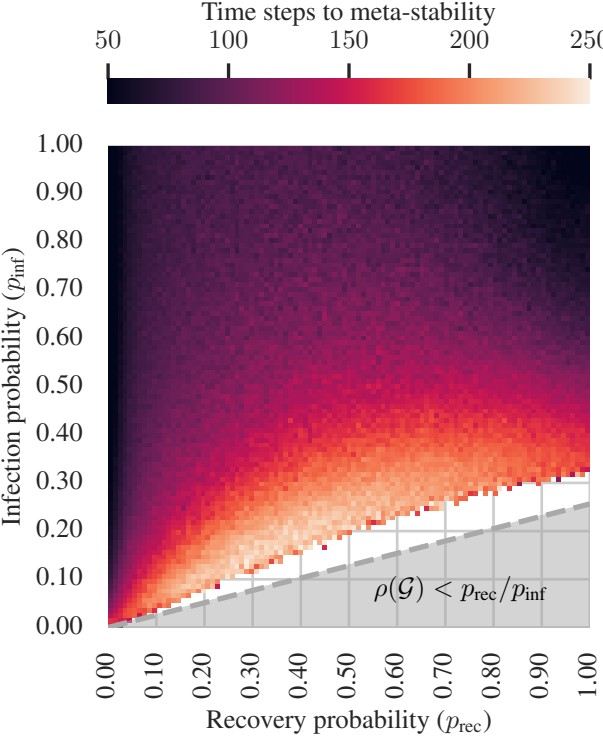

Figure 1: Heatmap of the average time steps for an SIS process $\{Y^{(t)}\}$ to reach meta-stability on graphs from the augmented `chn.2009.flu.1.00` dataset ($n = 40$, $\rho(\mathcal{G}) \approx 3.9$) as a function of the infection ($p_{\text{inf}}$) and recovery probability ($p_{\text{rec}}$). Meta-stability is determined using the convergence of the proportion of infected nodes. The white region at the bottom with no points indicates that no process reached meta-stability. The shaded gray area indicates the region where theory predicts rapid extinction (Theorem 4.1). Averaged over 100 runs.

vector in a round and the state vector in the consecutive round, respectively.

Now, given any vectors $\psi_{S'} \in \{0,1\}^{|S'|}$ and $\psi_S \in \{0,1\}^{|S|}$, we can define the standard (unbiased) *estimator* of $\mathbb{P}\left(\Psi_{S'}^{(\cdot+1)} = \psi_{S'} | \Psi_S^{(\cdot)} = \psi_S\right)$ using the dataset $\mathcal{D}$ as

$$\hat{\mathbb{P}}(\Psi_{S'}^{(\cdot+1)} = \psi_{S'} | \Psi_S^{(\cdot)} = \psi_S) =$$
$$\frac{1}{I(\psi_S)} \sum_{t \in \mathcal{I}(\psi_S)} \mathbb{1}\left\{\Psi_{S'}^{(t+1)} = \psi_{S'}\right\},$$

where $\mathcal{I}(\psi_S) = \left\{t \in T_D : \Psi_S^{(t)} = \psi_S\right\}$ denotes the set of time indices in the dataset where the vertices in set $S$ take the values specified in vector $\psi_S$, and $I(\psi_S) = |\mathcal{I}(\psi_S)|$. Subsequently, the estimators of DI and CI, denoted by $\hat{\mu}_{j|i}$ and $\hat{\nu}_{j|i,\psi_S}$, can be computed as described above.

## 3.3. Learning Algorithm and Guarantee

We now present our learning algorithm, called `SISLearn`, with pseudocode given in Algorithm 1. `SISLearn` takes as input the vertex set $V$, the dataset of infection state vectors $\mathcal{D}$, estimator error thresholds $\kappa_\mu$ and $\kappa_\nu$, infection probability $p_{\text{inf}}$, and the maximum degree $\Delta$ of the underlying graph. Note that if $p_{\text{inf}}$ is not known, it can be estimated from observational data, like done in Kirkeby et al. (2017) via a mean-field approximation. Additionally, we will show in Appendix A.3 that our algorithm is robust to misspecification of $p_{\text{inf}}$ and max degree $\Delta$.

The algorithm then learns the neighbors of each vertex $j$ via inclusion and exclusion. First, in lines 6-10, a superset of $\mathcal{N}(j)$, called $S$, is learned using Lemma 3.1. Next, in lines 11-15, `SISLearn` prunes $S$ by excluding non-neighbors using Corollary 3.1. Notice that Corollary 3.1 holds for any configuration $\psi_{S \setminus \{i\}}$, hence when applying it in Line 12, `SISLearn` selects the most data-rich configuration that maximizes the minimum number of samples available for estimating the two conditional probabilities that constitute $\nu_{j|i,S \setminus \{i\}}$. Formally, for a given vertex pair $(i, j)$ and candidate neighbor set $S \supseteq \mathcal{N}(j)$ for $j$, we define this configuration as $\psi_S^*(i, j) :=$

$$\underset{\psi \in \{0,1\}^{|S|-1}}{\arg\max} \min \left( I(\Psi_j^{(t)} = 0, \Psi_i^{(t)} = 1, \Psi_{S \setminus \{i\}}^{(t)} = \psi), \right.$$
$$\left. I(\Psi_j^{(t)} = 0, \Psi_i^{(t)} = 0, \Psi_{S \setminus \{i\}}^{(t)} = \psi) \right).$$

When clear from context, like in Line 12, we omit the dependence on $i$ and $j$, and simply write $\psi_S^*$.

Now, we state the learning guarantee for `SISLearn` that combines a bound on the sample complexity with the correctness of the algorithm.

**Theorem 3.1.** *(Learning Guarantee.)* Let $\varepsilon_\mu, \varepsilon_\nu, \zeta$ be positive constants, with $\varepsilon_\nu < p_{inf}(1-p_{inf})^{\Delta-1}/2$ and $\zeta \leq 1$. Suppose Assumption 3.1 holds. Further, assume that for all $i \neq j \in V$, all $S \subseteq V \setminus \{i, j\}$, and all $\psi_i, \psi_j \in \{0, 1\}$, and the respective $\psi_S^*(i, j)$, the following inequality holds:

$$I(\Psi_i = \psi_i, \Psi_j = \psi_j, \Psi_S = \psi_S^*(i, j)) \geq$$
$$\frac{2}{(1-\theta)^2(\min\{\varepsilon_\nu/2, \varepsilon_\mu\})^2} \log\left(\frac{2n^2(1+2^{n-1})}{\zeta}\right)$$

*Then, with probability at least $1 - \zeta$, Algorithm 1, when run with thresholds $\kappa_\nu = \varepsilon_\nu$ and $\kappa_\mu = \varepsilon_\mu$, returns the correct edge set of the underlying graph: $E_\mathcal{D} = E$.*

**Remark 3.2.** *Note that $\theta$, called the Markov contraction coefficient, is a constant determined by the graph structure and the transition probabilities of the Markov chain $\{\Psi^{(t)}\}$. It satisfies $0 \leq \theta < 1$ and remains independent of the number of samples $T$ drawn from the process.*

**Algorithm 1** `SISLearn`: Learn SIS Graph from Data
---
1: **Input:** Vertex set $V$, data $\mathcal{D}$, thresholds $\kappa_\mu, \kappa_\nu$, infection probability $p_{\text{inf}}$, max degree $\Delta$
2: **Output:** Learned graph edges $E_\mathcal{D}$
3: $E_\mathcal{D} \leftarrow \emptyset$
4: **for** $j \in V$ **do**
5: $\quad S \leftarrow \emptyset$
6: $\quad$ **for** $i \in V$ **do** $\qquad\qquad\qquad$ ▷ Inclusion
7: $\qquad$ **if** $\hat{\mu}_{j|i} \geq p_{\text{inf}} - \kappa_\mu$ **then**
8: $\qquad\quad S \leftarrow S \cup \{i\}$
9: $\qquad$ **end if**
10: $\quad$ **end for**
11: $\quad$ **for** $i \in S$ **do** $\qquad\qquad\qquad$ ▷ Exclusion
12: $\qquad$ **if** $\hat{\nu}_{j|i,\psi_S^*} < p_{\text{inf}}(1 - p_{\text{inf}})^{\Delta - 1} - \kappa_\nu$ **then**
13: $\qquad\quad S \leftarrow S \setminus \{i\}$
14: $\qquad$ **end if**
15: $\quad$ **end for**
16: $\quad E_\mathcal{D} \leftarrow E_\mathcal{D} \cup \{\{j,i\} : i \in S\}$
17: **end for**
18: **return** $E_\mathcal{D}$

## 4. Vaccination Strategies

Assuming we have learned the graph using `SISLearn`, we present two vaccination strategies inspired by the surrogate problem of *Spectral Radius Minimization* (SRM). We first establish a theoretical result relating the extinction time of the SIS model to the spectral radius of the graph and the SRM problem.

### 4.1. Extinction Time

Below, we formally state (and provide a proof in Appendix D.4.1) a sufficient condition for logarithmic extinction time that relates extinction time to the spectral radius of the graph. This result was first empirically established by Wang et al. (2003), and proven for this model by Ruhi et al. (2016).

**Theorem 4.1.** *The expected extinction time* $\mathbb{E}[\tau]$ *is upper bounded by* $\mathcal{O}(\log n)$ *if* $\rho(\mathcal{G}) < p_{rec}/p_{inf}$.

Theorem 4.1 establishes that rapid extinction hinges on the spectral radius of the graph, which leads to the natural approach of reducing the graph's spectral radius through vaccination or quarantining, modeled as the removal of vertices or edges. (Van Mieghem et al., 2011; Saha et al., 2015; Kiji et al., 2022). Building on this established approach, we focus on the surrogate SRM problem through vertex removals. It is important to note that our setting remains general: vaccinating a node does *not* remove it entirely unless $\alpha = 0$. As will be demonstrated in Section 5, our vaccination strategies are highly effective in reducing the extinction time, further justifying the use of the spectral radius as a proxy for the extinction time.

**Spectral Radius Minimization Problem.** Given a graph $\mathcal{G} = (V, E)$ with $n$ vertices and a budget $K$, the SRM problem aims to find a subset of at most $K$ vertices $R \subseteq V$ whose removal minimizes the spectral radius. Formally, the SRM problem solves

$$R^* = \operatorname*{arg\,min}_{R \subseteq V, |R| \leq K} \rho(\mathcal{G}[V \setminus R]). \qquad (1)$$

While the SRM problem is known to be NP-hard in general (Van Mieghem et al., 2011), in what follows we present a polynomial-in-$n$ optimal algorithm for graphs of bounded treewidth and a fast greedy heuristic for general graphs. In Appendix B, we provide an optimal algorithm for trees[2].

### 4.2. Vaccination Strategy for Graphs

Here, we propose a vaccination strategy that is an optimal algorithm for the SRM problem. Our approach consists of a dynamic programming (DP) procedure performed on a tree decomposition of the graph, allowing us to prove polynomial-in-$n$ time complexity for graphs of bounded treewidth. For readers unfamiliar with tree decomposition techniques, we detail a simpler optimal algorithm for trees in Appendix B, which shares the core ideas of our main algorithm. Additionally, we provide a primer on tree decompositions in Appendix C to facilitate understanding.

**Algorithm Overview.** Our algorithm, pseudocode given in Algorithm 2, employs a binary search strategy to identify the smallest possible spectral radius $\lambda_\varepsilon$ achievable by removing at most $K$ vertices from a graph $\mathcal{G}$, up to a precision of $\varepsilon > 0$. The binary search is performed on $[0, \Delta]$, as $\Delta$, the max degree of $\mathcal{G}$, is a trivial upper bound for spectral radius (Van Mieghem et al., 2011).

For each candidate value of $\lambda$ during the binary search, we invoke a feasibility check algorithm, pseudocode given in Algorithm 3, to determine whether there exists a set of at most $K$ vertices whose removal ensures that the spectral radius of the resulting graph does not exceed $\lambda$. Below, we detail the feasibility check algorithm.

Given a graph $\mathcal{G} = (V, E)$ with treewidth $\text{tw}(\mathcal{G}) \leq \omega$, we first decompose the graph into a *nice tree decomposition* $\mathcal{T} = (T, \{X_t\}_{t \in V(T)})$, where $T$ is a tree whose every node $t$ is assigned a bag $X_t \subseteq V$. To minimize ambiguity, we use vertex (vertices) when referring to elements of the graph $\mathcal{G}$ and node (nodes) when referring to elements of the tree decomposition $\mathcal{T}$.

We then perform a bottom-up dynamic programming algorithm on the tree decomposition to find the optimal vaccination set for each node $t$ of the tree. First, define $V_t$ as

---

[2]Note that the SRM problem is NP-hard for general graphs and thus there exists no polynomial-time exact algorithm, unless P = NP.

the union of bags in the subtree $\mathcal{T}$ rooted at $t$, including $X_t$. Then, for every node $t \in V(\mathcal{T})$, every $S \subseteq X_t$, and all $c \in \{0, \ldots, K\}$ define $\mathrm{DP}[t, S, c]$ as

$$\mathrm{DP}[t, S, c] =$$
$$\left\{R \subseteq V_t \colon |R| = c, S \cap R = \emptyset, \rho(G[V_t \setminus R]) \le \lambda\right\}.$$

In words, $\mathrm{DP}[t, S, c]$ stores the set of feasible removal sets of size $c$ from $V_t$ such that $S$ is preserved and the spectral radius of the induced subgraph is at most $\lambda$. Notice that if $r \in V(\mathcal{T})$ is the root of $\mathcal{T}$, then $R \in \mathrm{DP}[r, \emptyset, c]$ is a feasible removal set with $c$ elements that satisfies $\rho(\mathcal{G}[V(\mathcal{G}) \setminus R]) \le \lambda$. If no such set exists (i.e., $\mathrm{DP}[r, \emptyset, c] = \emptyset$), then there exists no feasible removal set of size at most $c$.

In what follows, we describe how Algorithm 3 populates $\mathrm{DP}[t, \cdot, \cdot]$, initialized to $\emptyset$, for each node $t$ of the tree in post-order, depending on the node's type (refer to Appendix C for types).

**Leaf node**: If $t$ is a leaf node, then $X_t = \emptyset$ and $\mathrm{DP}[t, \emptyset, 0] = \{\emptyset\}$.

**Introduce node**: If $t$ is an introduce node with child node $t'$, then, $X_t = X_{t'} \cup \{v\}$ for some $v \in V(\mathcal{G})$. For every $S \subseteq X_t$ and for all $1 \le c \le K$, if $v \notin S$, then we set $\mathrm{DP}[t, S, c] = \{R' \cup \{v\} \colon R' \in \mathrm{DP}[t', S, c-1]\}$.

If $v \in S$, then we need to verify if $\rho(G[V_t \setminus R']) \le \lambda$ for each $R' \in \mathrm{DP}[t', S \setminus \{v\}, c]$. So, $\mathrm{DP}[t, S, c] = \{R' \in \mathrm{DP}[t', S \setminus \{v\}, c] \colon \rho(G[V_t \setminus R']) \le \lambda\}$.

**Forget node**: If $t$ is a forget node with child node $t'$, then, $X_t = X_{t'} \setminus \{w\}$ for some $w \in V(\mathcal{G})$ and thus $V_t = V_{t'}$. Then, for every $S \subseteq X_t$ and $0 \le c \le K$, set $\mathrm{DP}[t, S, c] = \mathrm{DP}[t', S, c] \cup \mathrm{DP}[t', S \cup \{w\}, c]$.

**Join node**: If $t$ is a join node with two children $t_1$ and $t_2$, then $X_t = X_{t_1} = X_{t_2}$. Then, for every $S \subseteq X_t$, $0 \le c_1, c_2 \le K$, and $R_1 \in \mathrm{DP}[t_1, S, c_1], R_2 \in \mathrm{DP}[t_2, S, c_2]$, define $R = R_1 \cup R_2$. Then, iteratively set $\mathrm{DP}[t, S, c] = \mathrm{DP}[t, S, c] \cup \{R\}$ if $c = |R| \le K$ and $\rho(G[V_t \setminus R]) \le \lambda$.

**Computational Complexity.** We now state (and prove in Appendix D.4.3) the time complexity of Algorithm 2, showing that it is polynomial in the number of vertices $n$ for graphs of bounded treewidth $\omega$.

**Theorem 4.2.** *Given an input graph $\mathcal{G}$ with treewidth $\mathrm{tw}(\mathcal{G}) \le \omega$, budget $K$, and precision $\varepsilon$, Algorithm 2 has a worst-case time complexity of $\mathcal{O}\left(n^{\mathcal{O}(1)} K^{\mathcal{O}(1)} 2^{\mathcal{O}(\omega)} \log(\Delta/\varepsilon)\right)$.*

The time complexity of our algorithm aligns with that of other NP-hard problems, such as the maximum independent set and Hamiltonian path, which are solvable in polynomial-in-$n$ time on graphs with bounded treewidth using DP (Bodlaender et al., 2013). It should be noted that despite achiev-

---

**Algorithm 2** DP algorithm for vaccination

1: **Input:** Graph $\mathcal{G}$, treewidth upperbound $\mathrm{tw}(\mathcal{G}) \le \omega$, budget $K$, precision $\varepsilon > 0$
2: **Output:** Set of vertices to vaccinate $R_\varepsilon$
3: low $\leftarrow 0$
4: high $\leftarrow \Delta$          ▷ Use max. deg. as upper bound
5: $R_\varepsilon \leftarrow \emptyset$
6: $\mathcal{T} \leftarrow$ a nice tree decomposition of $\mathcal{G}$ with width $\le \omega$
7: **while** high $-$ low $> \varepsilon$ **do**
8:      mid $\leftarrow \frac{\mathrm{low+high}}{2}$
9:      (feasible, $\hat{R}$) $\leftarrow$ ALGORITHM 3($\mathcal{G}, \mathcal{T}, K, \mathrm{mid}$)
10:      **if** feasible **then**
11:          high $\leftarrow$ mid
12:          $R_\varepsilon \leftarrow R$
13:      **else**
14:          low $\leftarrow$ mid
15:      **end if**
16: **end while**
17: **Return** $R_\varepsilon$

---

**Algorithm 3** Feasibility checker for graph vaccination

1: **Input:** Graph $\mathcal{G}$, nice tree decomposition $\mathcal{T}$, budget $K$, threshold $\lambda$
2: **Output:** Feasibility indicator and vaccination set $R$
3: **Initialize** $\mathrm{DP}[t, S, c] \leftarrow \emptyset$ for all $t$ of $\mathcal{T}$, $S \subseteq X_t$, and $c \in [0, K]$
4: $\mathcal{T}_{\mathrm{po}} \leftarrow$ nodes of $\mathcal{T}$ in post-order
5: $r \leftarrow$ root of $\mathcal{T}$
6: **for** node $t \in \mathcal{T}_{\mathrm{po}}$ **do**
7:      **for** subset $S \subseteq X_t$ **do**
8:          **for** $c \in [0, K]$ **do**
9:              Update $\mathrm{DP}[t, S, c]$ based on node type
10:          **end for**
11:      **end for**
12: **end for**
13: **if** $\mathrm{DP}[r, \emptyset, c] = \emptyset$ for all $c \in [0, K]$ **then**
14:      **Return** (**False**, $\emptyset$)
15: **else**
16:      $c^* \leftarrow \min\{c \in [0, K] \colon \mathrm{DP}[r, \emptyset, c] \ne \emptyset\}$
17:      **Return** (**True**, any $R \in \mathrm{DP}[r, \emptyset, c^*]$)
18: **end if**

---

ing similar complexity, the decision version of the SRM problem cannot, to the best of our knowledge, be expressed in monadic second-order logic, rendering Courcelle's theorem[3] inapplicable (Courcelle, 1990). Consequently, it was previously unknown whether a polynomial-in-$n$ time algorithm for SRM on bounded treewidth graphs existed.

---

[3]Informally, Courcelle's theorem states that any graph decision problem expressible in monadic second-order logic can be decided in linear time for graphs of bounded treewidth. We refer the reader to Chapter 7.4 of Cygan et al. (2015) for more details.

---

**Algorithm 4** Greedy vaccination algorithm

---

1: **Input:** Graph $\mathcal{G}$, budget $K$
2: **Output:** Set of nodes $R$ to vaccinate
3: $R \leftarrow \emptyset$
4: $\mathcal{G}' \leftarrow \mathcal{G}$
5: **while** $|R| < K$ **do**
6: $\quad v^* \leftarrow \arg\min_{v \in V(\mathcal{G}')} \rho(\mathcal{G}'[V(\mathcal{G}') \setminus \{v\}])$
7: $\quad R \leftarrow R \cup \{v^*\}$
8: $\quad \mathcal{G}' \leftarrow \mathcal{G}'[V(\mathcal{G}') \setminus \{v^*\}]$
9: **end while**
10: **Return** $R$

---

### 4.3. Heuristic Vaccination Strategy

Algorithm 2 exactly solves the NP-hard SRM problem as a surrogate for the vaccination problem. Although this algorithm is optimal and achieves the lowest spectral radius possible for a given budget and thus the lowest extinction time, as will be verified in the experiments, it is computationally expensive for general graphs with unknown treewidth. To address this, we also propose a heuristic algorithm that is not only computationally efficient but also performs well in practice.

**Algorithm Overview.** Algorithm 4 iteratively removes the vertex that reduces the spectral radius the most.

**Computational Complexity.** The worst-case complexity of the algorithm is $\mathcal{O}\left(Kn^3\right)$. The complexity arises from (i) $K$ iterations in the outer loop; (ii) the spectral radius computation, which has a complexity of $\mathcal{O}(n^2)$ using the Lanczos algorithm for sparse graphs with $\mathcal{O}(n)$ nonzero entries (Cullum & Donath, 1974); and (iii) sorting the vertices based on the reduction in spectral radius, which has a complexity of $\mathcal{O}(n)$.

### 4.4. Combined Learning and Vaccination

To solve the VUG problem, we first use `SISLearn` to infer the graph from data. Subsequently, we utilize the learned graph with either Algorithm 2 or Algorithm 4 to select $K$ vertices for vaccination, depending on the graph's treewidth. Specifically, as supported by the empirical results in Appendices A.8 and A.9, `DP` Algorithm 2 is computationally feasible for graphs with treewidth up to 15, enabling exact vaccination. For larger treewidths, we employ `Greedy` Algorithm 4.

## 5. Experiments

In this section, we present experiments using real-world graphs to evaluate the performance of our combined learning and vaccination approach on the VUG problem. A comprehensive set of experiments analyzing the perfor-

mance of our learning and vaccination algorithms separately are presented in Appendix A. We provide the implementation of our algorithms as well as all baselines at https://github.com/sepehr78/learn2vac.

**Setting.** We consider a contagion spreading on a real-world spreading network from OutbreakTrees (Taube et al., 2022), a database of infectious disease transmission trees compiled from real-world disease outbreaks. We utilize the `chn.2009.flu.1.00` dataset, representing the transmission tree of the 2009 influenza A outbreak in Beijing, China, comprising $n = 40$ vertices. To simulate variability, we augment the tree by adding edges between unconnected vertex pairs independently with a probability of $0.05$, generating $40$ distinct random graphs, with an average treewidth of $8$.

We simulate the SIS model on these graphs with parameters: $p_{\text{init}} = 0.3$, $p_{\text{inf}} = 0.3$, and $p_{\text{rec}} = 0.5$. The vaccination budget is set to $K = 7$ with $\alpha = 0.2$, reflecting the $80\%$ efficacy of the 2009 Beijing influenza vaccine (Wu et al., 2010). For each graph, we generate $40$ random initial infection states and run the simulation for $T = 2000$ rounds.

**Algorithms.** Our approach involves two main steps: (1) learning the graph from the observed infection data of the first $T/2$ rounds using Algorithm 1, and (2) applying our vaccination algorithms, `DP` (Algorithm 2) and `Greedy` (Algorithm 4) to the learned graph.

It should be noted that we attempted to compare `SISLearn` with the MLE approach of Barbillon et al. (2020), the only work to-date on SIS structure learning, but they assume each vertex gets infected by only one neighbor at a time. This assumption enforces a linear relationship between the number of infected neighbors and the probability of a vertex becoming infected. In our model, all infected neighbors attempt to infect a susceptible vertex simultaneously, leading to an overall higher infection probability. This causes their method to always return a complete graph in our more general setting, making meaningful comparison infeasible.

For comparison, we evaluate the following baseline vaccination strategies on the learned graph:

- `Rand.`: vaccinate $K$ random vertices.
- `LD`: vaccinate $K$ vertices with the largest degree.
- `PO` (Scaman et al., 2016): vaccinate $K$ vertices at the front of the priority order.
- `GW` (Saha et al., 2015): vaccinate $K$ vertices appearing in the most number of closed walks.

The hyperparameters required by the above methods have been set as prescribed in the original papers. We set $\kappa_\mu = \kappa_\nu = 0.01$ for Algorithm 1 and use the positive-instance driven approach of Tamaki (2019) to compute the tree decomposition for Algorithm 2.

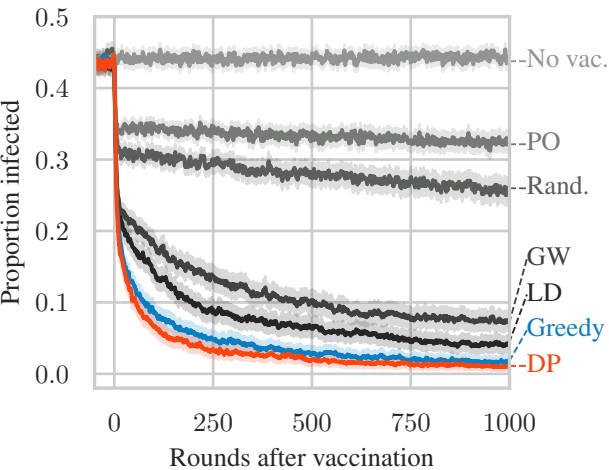

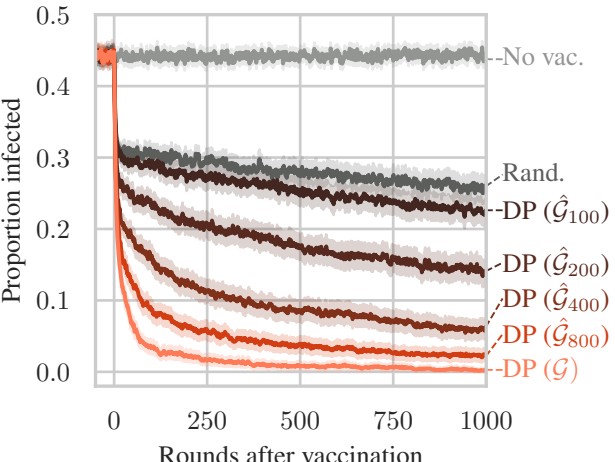

Figure 2: Proportion of infected nodes versus rounds after vaccination using different strategies on learned graph from `SISLearn`. Shaded regions represent 99% confidence interval.

Figure 3: Proportion of infected nodes versus rounds after vaccination via Algorithm 2 on the learned graph ($\hat{\mathcal{G}}_{T'}$) from `SISLearn` using different numbers of rounds ($T'$) for learning. Shaded regions represent 99% confidence interval.

**Results.** Figure 2 presents the average proportion of infected nodes over all runs against rounds following vaccination, comparing different vaccination strategies applied to the learned graph. Notably, all non-random vaccination methods—except `PO`—outperform random vaccination, demonstrating that our learning algorithm has effectively learned the graph. Our proposed algorithms, `DP` and `Greedy`, consistently outperform all baselines. Specifically, `DP` achieves the lowest infection rates across all rounds, with `Greedy` closely trailing behind. `LD` ranks third, demonstrating that while targeting high-degree nodes is effective, it does not match the performance of our algorithms. Notably, the `PO` strategy performs the worst, even underperforming the random vaccination baseline. This underperformance is likely because `PO` is designed for 'healing' vertices in continuous-time SIS models (Scaman et al., 2016), making it less suitable for our discrete-time simulation.

**Robustness Analysis.** To assess the robustness of our combined approach, we vary the number of learning rounds used by `SISLearn` before applying Algorithm 2 for vaccination. We experiment with learning rounds ranging from 100 to 800 and illustrate the results in Figure 3. Our approach consistently outperforms random vaccination even with as few as 100 learning rounds. Performance improves significantly as the number of learning rounds increases, eventually plateauing around 800 rounds. This indicates that our method remains effective across varying amounts of learning data. We provide additional robustness analysis using the other vaccination algorithms in Appendix A.4.

## 6. Conclusion

We introduced the VUG problem, combining structure learning for SIS processes with strategic vaccination when the underlying contact network is unknown. Our framework combines a novel inclusion-exclusion–based learning algorithm—supported by sample-complexity guarantees—with vaccination strategies aimed at minimizing the graph's spectral radius. In particular, the proposed dynamic programming approach achieves optimal results on graphs of bounded treewidth, while a simpler greedy heuristic scales effectively to general networks. Experimental results on real-world data confirm that our method significantly accelerates epidemic extinction and reduces infection rates, even when limited observations are available.

For future work, a promising direction involves examining which aspects of the network structure are learned first, allowing us to optimize the balance between prolonged learning and timely vaccinations. Furthermore, establishing theoretical approximation guarantees for vaccination heuristics remains an open problem.

## Acknowledgments

This work was partially supported by the SNF project 200021_204355 / 1, *Causal Reasoning Beyond Markov Equivalencies*.

## Impact Statement

This paper presents theoretical advances in machine learning for epidemic modeling and outbreak control, with potential applications in public health. We defer discussion of broader societal consequences to domain-specific implementations of our framework.

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

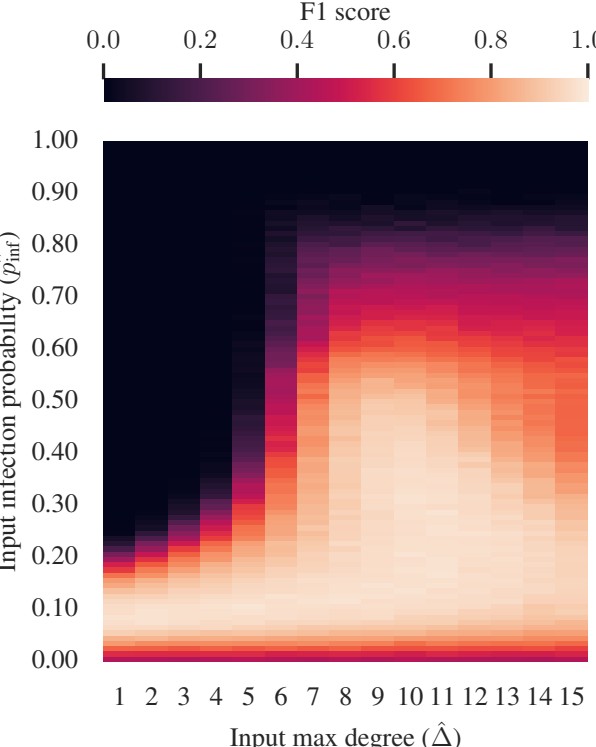

Figure 4: Heatmap of the average F1 score of `SISLearn` with 1000 rounds of learning on the augmented `chn.2009.flu.1.00` dataset (true $p_{\text{inf}} = 0.3$ and true $\Delta = 10$) as a function of the input infection probability ($\hat{p}_{\text{inf}}$) and input max degree ($\hat{\Delta}$). Averaged over 100 runs.

## A. Additional Experiments

In this appendix, we provide additional extensive experiments on the performance and robustness of our learning and vaccination algorithms. All experiments were conducted on a machine equipped with two Intel Xeon E5-2680 v3 CPUs, 256GB of RAM, and running Ubuntu 24.04.1 LTS. We used Python 3.11.11 and NetworkX 3.4.2 for graph manipulation and generation.

### A.1. Time to Reach Meta-Stability

To investigate the time required for the SIS process to reach a meta-stable state, we conducted simulations using the same augmented `chn.2009.flu.1.00` graphs from the main experiments (Section 5). For these simulations, we systematically varied the infection probability ($p_{\text{inf}}$) and the recovery probability ($p_{\text{rec}}$) from 0 to 1. Meta-stability was determined to be reached when the proportion of infected nodes over a moving window of recent time steps converged, indicating that the overall infection level had stabilized. We recorded the number of rounds (time steps) until this convergence criterion was met.

**Results.** Figure 1 displays a heatmap of the average number of time steps required for the SIS process to reach meta-stability, as a function of the infection probability ($p_{\text{inf}}$) and recovery probability ($p_{\text{rec}}$). The average spectral radius of the graphs was $\rho(\mathcal{G}) \approx 3.9$. The results demonstrate that, for a wide range of parameter settings where the infection persists (i.e., does not quickly go extinct, indicated by the white region and the theoretically predicted extinction region shown in gray), meta-stability is achieved relatively quickly. Even in the slowest-converging scenarios (brighter regions in the heatmap), meta-stability is reached within 250 rounds. This rapid convergence to a quasi-stationary state provides empirical support for Assumption 3.1.

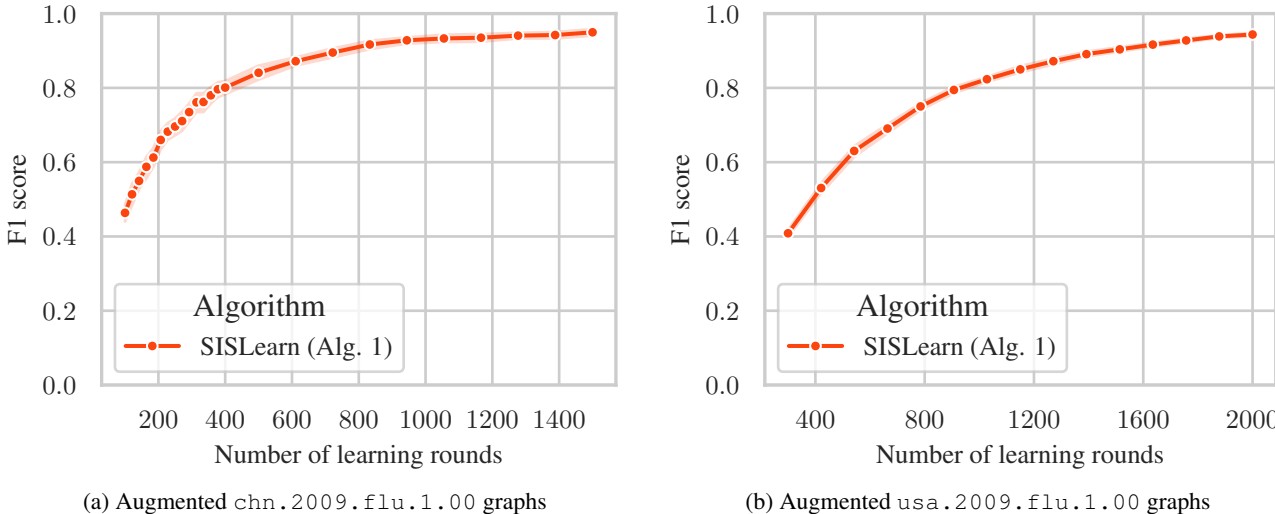

(a) Augmented `chn.2009.flu.1.00` graphs        (b) Augmented `usa.2009.flu.1.00` graphs

Figure 5: Average F1-score of the learned graph from `SISLearn` versus number of learning rounds used on the augmented `chn.2009.flu.1.00` and `usa.2009.flu.1.00` graphs. Shaded regions represent the $99\%$ confidence interval.

### A.2. Learning Performance of `SISLearn`

We investigate the performance of `SISLearn` in learning the underlying graph of the SIS epidemic on real-world outbreak graphs. We consider two settings:

**2009 flu outbreak in Beijing, China.** This is the same setting as the main experiments, using the `chn.2009.flu.1.00` network with $n = 40$ vertices, augmented with $5\%$ additional edges, to get a total of 40 graphs. We simulate the SIS model on these graphs with parameters: initial infection probability $p_{\text{init}} = 0.3$, infection probability $p_{\text{inf}} = 0.3$, and recovery probability $p_{\text{rec}} = 0.5$

**2009 flu outbreak in Pennsylvania, USA.** For this setting, we use the much larger `usa.2009.flu.1.00` contact network, from the 2009 H1N1 influenza outbreak in Pennsylvania, USA, with $n = 286$ vertices (Taube et al., 2022). We augment the network with $1\%$ additional edges to get a total of 40 graphs. We simulate the SIS model on these graphs with parameters: initial infection probability $p_{\text{init}} = 0.3$, infection probability $p_{\text{inf}} = 0.2$, and recovery probability $p_{\text{rec}} = 0.5$.

**Results.** We present the average F1-score of the learned graph from `SISLearn` versus the number of learning rounds used in Figure 5. We observe that for both settings, the F1-score increases rapidly with the number of learning rounds at the start and eventually plateauing. Importantly, `SISLearn` is able to achieve a high F1-score ($> 0.9$) for both graphs with sufficient number of learning rounds.

### A.3. Robustness of `SISLearn` to Input Misspecification

We evaluate the robustness of `SISLearn` to misspecification of its input parameters: the infection probability ($p_{\text{inf}}$) and the maximum degree ($\Delta$). The experimental setup mirrors that described in Section 5, using the augmented `chn.2009.flu.1.00` dataset. For data generation, the true SIS model parameters were an infection probability $p_{\text{inf}} = 0.3$ and an actual maximum degree $\Delta = 10$. `SISLearn` was then executed for 1000 learning rounds, with systematically varied inputs for $\hat{p}_{\text{inf}}$ (ranging from 0.01 to 1.00) and $\hat{\Delta}$ (ranging from 1 to 15). After learning, we computed the F1 score of graph recovery, averaged over 100 independent simulation runs.

**Results.** Figure 4 presents a heatmap of the average F1 score achieved by `SISLearn` as a function of the input infection probability $\hat{p}_{\text{inf}}$ ($y$-axis) and the input maximum degree $\hat{\Delta}$ ($x$-axis). The heatmap clearly demonstrates that `SISLearn` exhibits considerable robustness to variations in these input parameters. Optimal performance, indicated by F1 scores approaching 1.0 (brightest regions), is observed when $\hat{p}_{\text{inf}}$ is in the vicinity of the true value (e.g., approximately 0.2 to 0.4) and $\hat{\Delta}$ is sufficiently large (e.g., $\hat{\Delta} \geq 7$). Importantly, `SISLearn` maintains high F1 scores (often exceeding 0.8)

even when $\hat{p}_{\text{inf}}$ is moderately overestimated (e.g., up to $0.5 - 0.6$) or when $\hat{\Delta}$ is underestimated (e.g., down to $5 - 6$). This robustness suggests that precise prior knowledge of $p_{\text{inf}}$ and $\Delta$ is not a stringent requirement for achieving effective graph recovery with `SISLearn`.

### A.4. Performance of `SISLearn` with Different Vaccination Strategies

Here, we present results on the robustness of our learning algorithm combined with other vaccination strategies. We use the same setting as in Section 5 and vary the number of learning rounds from 100 to 800. We then apply separately apply `Greedy`, `LD`, and `GW` for vaccination.

**Results.** In Figure 6, we present the average fraction of infected nodes over time following vaccination using the different algorithms on the learned graph. We observe that `Greedy` is the only algorithm that outperforms random with only 100 rounds of learning. `Greedy` also performs the best with the most learning rounds, followed by `LD` and `GW`. Interestingly, `LD` performs better than `Greedy` when $T' = 200$ and $T' = 400$, potentially indicating that our learning algorithm recovers high-degree vertices quickly.

### A.5. Performance of Algorithm 2 and Algorithm 4

Here, we present results on the effectiveness of our proposed vaccination strategies, Algorithm 2 and Algorithm 4, on the real-world outbreak graphs. Unlike in the main experiments, we do not used the learned graph and instead apply our algorithms and baselines directly on the original graphs.

**2009 flu outbreak in Beijing, China.** This is the same setting as the main experiments, using the `chn.2009.flu.1.00` network with $n = 40$ vertices, augmented with $5\%$ additional edges, to get a total of 40 graphs. We simulate the SIS model on these graphs with parameters: initial infection probability $p_{\text{init}} = 0.3$, infection probability $p_{\text{inf}} = 0.3$, and recovery probability $p_{\text{rec}} = 0.5$. We set the vaccination budget to $K = 7$ with an efficacy of $80\%$ ($\alpha = 0.2$).

**2009 flu outbreak in Pennsylvania, USA.** For this setting, we use the much larger `usa.2009.flu.1.00` contact network, from the 2009 H1N1 influenza outbreak in Pennsylvania, USA, with $n = 286$ vertices. We augment the network with $1\%$ additional edges to get a total of 40 graphs, with an average treewidth of 58. We simulate the SIS model on these graphs with initial infection probability $p_{\text{init}} = 0.3$, infection probability $p_{\text{inf}} = 0.2$, and recovery probability $p_{\text{rec}} = 0.5$. We set the vaccination budget to $K = 50$ with an efficacy of $75\%$ ($\alpha = 0.25$), reflecting the efficacy of the influenza vaccine in the US in 2009 (Borse et al., 2013). As the treewidths are much larger than the `chn.2009.flu.1.00` dataset (58 versus 8), computing the tree decomposition for Algorithm 2 is computationally very expensive, and thus we only present results for Algorithm 4. Similarly, computing the priority order for `PO` is also infeasible as it is an NP-hard problem (Scaman et al., 2016), and thus we do not present results for this method.

**Baselines.** For convenience we restate the baselines as used in the main paper:

- `Rand.`: vaccinate $K$ random vertices.
- `LD`: vaccinate $K$ vertices with the largest degree.
- `PO` (Scaman et al., 2016): vaccinate $K$ vertices at the front of the priority order.
- `GW` (Saha et al., 2015): vaccinate $K$ vertices appearing in the most number of closed walks.

**Results.** We present the average proportion of infected nodes over time following vaccination using different strategies on the augmented `chn.2009.flu.1.00` and `usa.2009.flu.1.00` graphs in Figure 7. We observe that Algorithm 2 consistently outperforms all baselines, followed by `Greedy` and `LD` (see Figure 7a). On the larger `usa.2009.flu.1.00` graph in Figure 7b, `Greedy` performs the best, followed by `LD` and `GW`.

### A.6. Impact of Epidemic Severity on Vaccination

We further evaluate our combined learning and vaccination approach under varying conditions of epidemic severity and available vaccination resources. The base experimental setting is identical to that described in Section 5 (augmented `chn.2009.flu.1.00` dataset, `SISLearn` for graph inference, followed by vaccination). We consider two specific scenarios:

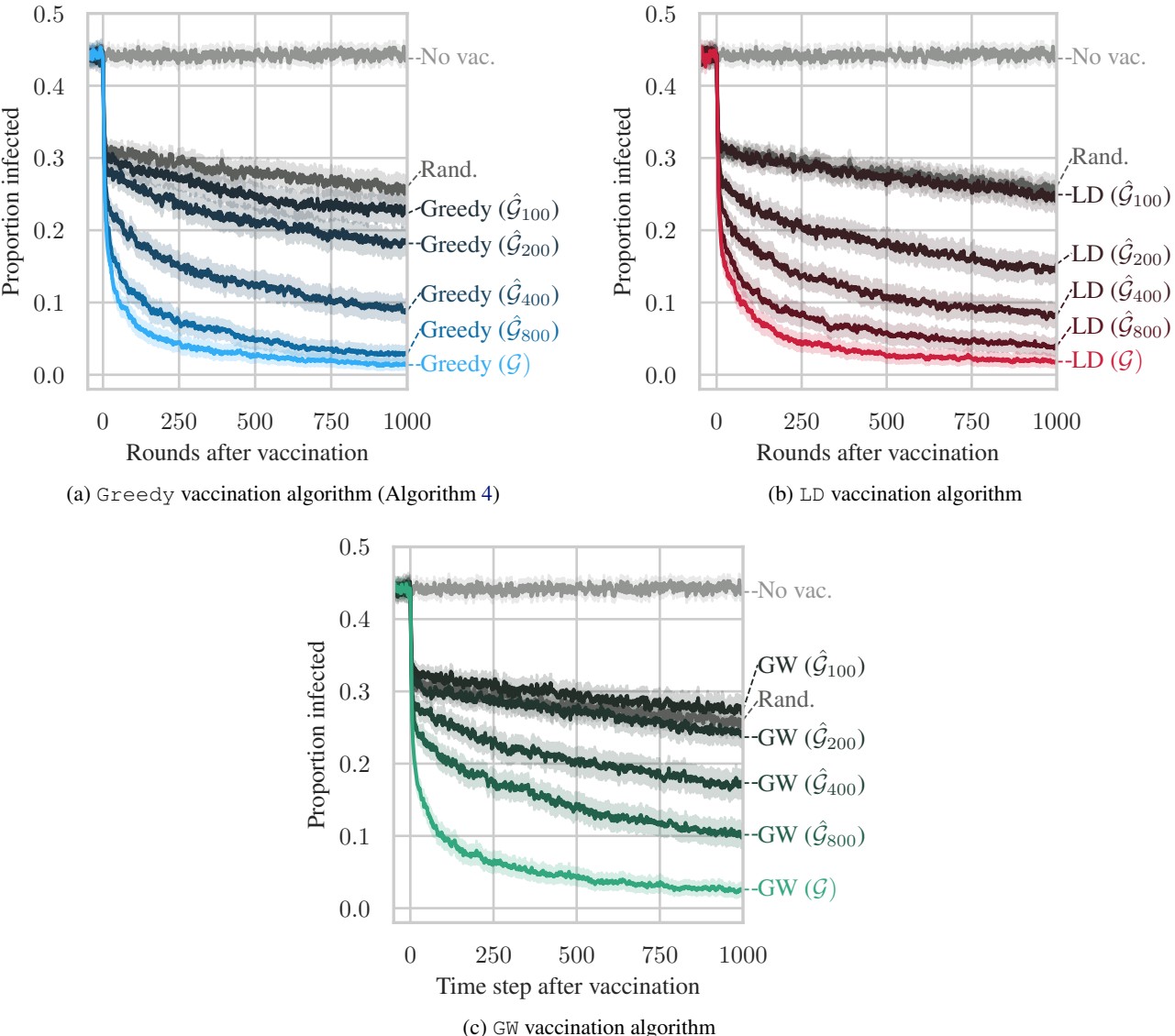

Figure 6: Proportion of infected nodes versus rounds after vaccination using different algorithms on the learned graph ($\hat{\mathcal{G}}_{T'}$) from SISLearn with varying numbers of learning rounds ($T'$). Shaded regions represent a 99% confidence interval.

- **Low Severity / Low Budget:** The SIS model is simulated with a lower infection probability $p_{\mathrm{inf}} = 0.15$. After learning the graph from the first $T/2$ rounds of this simulation, vaccination strategies are applied with a reduced budget of $K = 3$.
- **High Severity / High Budget:** The SIS model is simulated with a higher infection probability $p_{\mathrm{inf}} = 0.80$ and an increased budget of $K = 25$.

All other parameters, including recovery probability $p_{\mathrm{rec}} = 0.5$, vaccination efficacy $\alpha = 0.2$, and learning parameters for SISLearn, remain consistent with those in Section 5. Results are averaged over 40 runs.

**Results.** Figure 8 illustrates the proportion of infected nodes following vaccination for the two described scenarios.

In the low severity scenario with a low budget (Figure 8a), the epidemic is inherently less aggressive. Even without vaccination, the infection level is considerably lower than in the main experiment. With a small budget of $K = 3$, our DP and Greedy strategies, applied to the graph learned by SISLearn, rapidly reduce the infection to near extinction. While

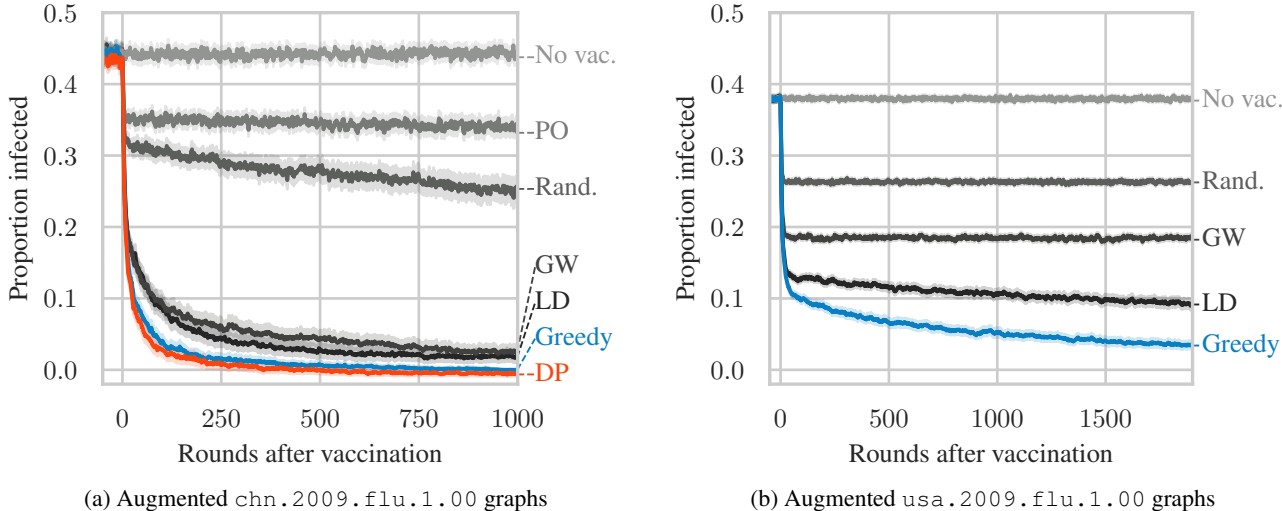

(a) Augmented `chn.2009.flu.1.00` graphs       (b) Augmented `usa.2009.flu.1.00` graphs

Figure 7: Proportion of infected nodes versus rounds after vaccination using different strategies on the augmented `chn.2009.flu.1.00` and `usa.2009.flu.1.00` graphs. Shaded regions represent the 99% confidence interval.

other strategies also show benefit, the targeted approaches achieve the most significant reduction, effectively controlling the mild outbreak with minimal resources.

Conversely, in the high severity scenario with a high budget (Figure 8b), the "No vaccination" case shows a very high persistent infection level. Here, a substantial vaccination effort is required. Our `DP` and `Greedy` algorithms again demonstrate superior performance, achieving a much more significant reduction in infection compared to baseline heuristics like `LD` and `GW`. This highlights the importance of optimized vaccination strategies when facing highly contagious outbreaks.

### A.7. Empirical Running Time of `SISLearn`

Here, we empirically demonstrate that our `SISLearn` algorithm is computationally efficient and can scale to very large graphs. We generate connected Erdős-Rényi (ER) graphs with $n \in \{1000, \ldots, 10000\}$ vertices and edge probability $p = 1.1 \log n/n$. We set the number of learning rounds to $T = 2000$ and run `SISLearn` to learn the underlying graph, measuring its running time. For each $n$, we generate 10 random graphs and report the average running time.

**Results.** Figure 9 shows the average running time of `SISLearn` across different number of vertices. As shown, `SISLearn` can easily handle graphs as large as $n = 10000$ vertices and $\approx 50000$ edges, taking only 9 minutes to run, demonstrating that `SISLearn` is computationally efficient and can scale to very large graphs.

### A.8. Empirical Running Time of `DP`

In this section, we empirically demonstrate the superior performance of our `DP` Algorithm 2 in solving the SRM problem on graphs with bounded and unbounded treewidth, compared with an exhaustive search algorithm.

**Small Treewidth.** We generate graphs of treewidth 3 with $n \in \{10, \ldots, 98\}$ vertices, and set the budget $K = 0.2n$. We then run our `DP` algorithm and an exhaustive search algorithm, which checks all $\binom{n}{k}$ subsets, to find the optimal vaccination set (i.e., solve the SRM problem), measuring their running time. Note that both approaches are exact algorithms and return the optimal solution. For each $n$, we generate 40 random graphs and report the average running time.

**Unbounded Treewidth.** We generate connected Erdős-Rényi (ER) graphs with $n \in \{10, \ldots, 90\}$ vertices and edge probability $p = 1.1 \log n/n$, which results in graphs with unbounded treewidth. We also set the budget $K = 0.2n$ and run our `DP` algorithm and the exhaustive search algorithm to find the optimal vaccination set, measuring their running time. For each $n$, we also generate 40 random graphs and report the average running time.

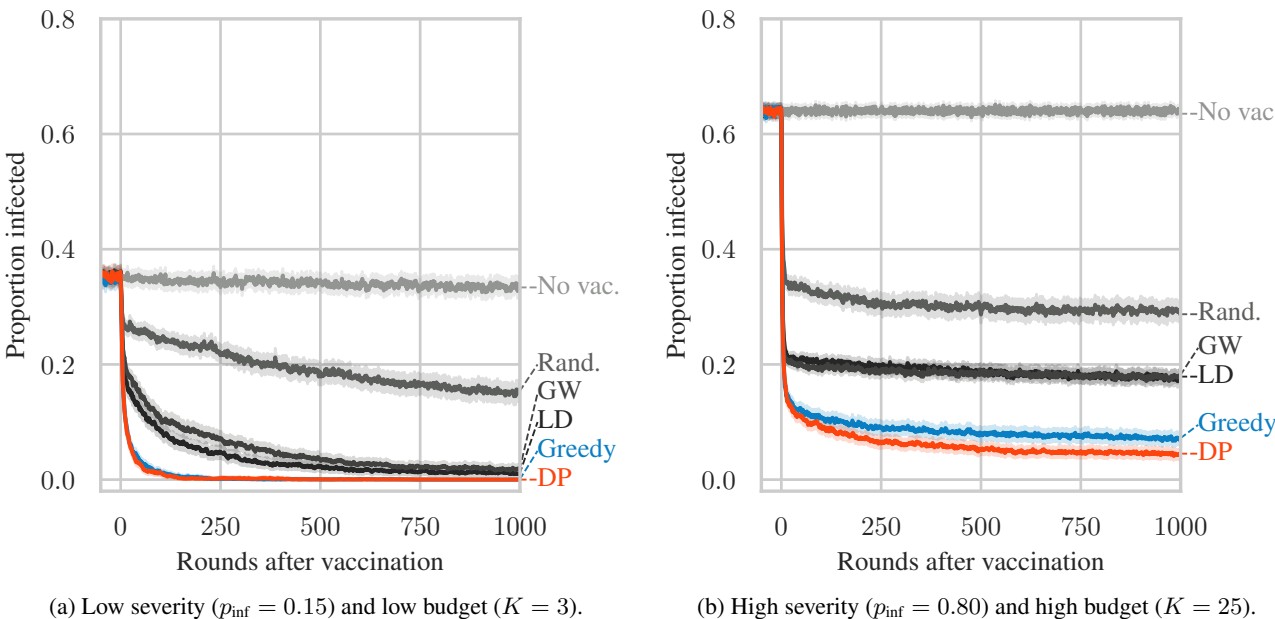

(a) Low severity ($p_{\text{inf}} = 0.15$) and low budget ($K = 3$).    (b) High severity ($p_{\text{inf}} = 0.80$) and high budget ($K = 25$).

Figure 8: Proportion of infected nodes versus rounds after vaccination using different strategies on the graph learned by `SISLearn` from the augmented `chn.2009.flu.1.00` dataset with low and high severity. Shaded regions represent the 99% confidence interval, averaged over 40 runs.

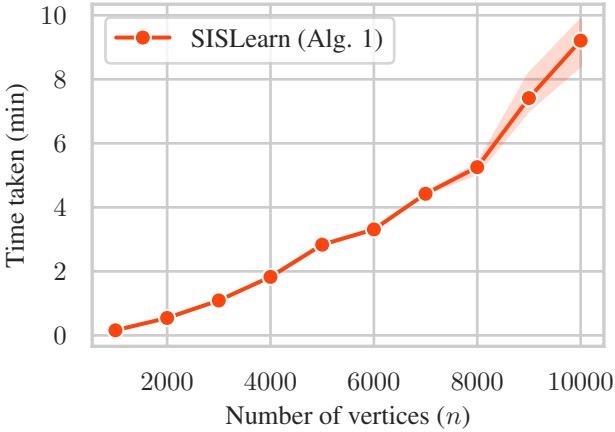

Figure 9: Running time (in minutes) of the `SISLearn` algorithm on connected ER graphs, plotted against the number of vertices. Shaded regions denote the 99% confidence interval, averaged over 10 runs.

**Large Treewidth.**    We also conduct additional experiments on large graphs with treewidth 10, for which we only run our `DP` algorithm, as the exhaustive search algorithm is computationally infeasible. We generate graphs with $n \in \{20, \dots, 2000\}$ vertices and set the budget $K = 0.2n$. For each $n$, we generate 40 random graphs and report the average running time.

**Results.**    In Figure 10, we compare the average running times of `DP` and the exhaustive search algorithm across the small and unbounded treewidth settings. As shown, `DP` is orders of magnitude faster than exhaustive search for $n \geq 20$. In the small treewidth setting (Figure 10a), the exhaustive search time grows exponentially with $n$, while `DP` exhibits polynomial growth, consistent with the theoretical guarantee in Theorem 4.2. In the unbounded treewidth setting (Figure 10b), `DP` also incurs exponential time, which is to be expected for the NP-hard SRM problem, but remains substantially faster than exhaustive search.

Figure 11 shows the average running time of `DP` on graphs with treewidth 10. As shown, `DP` exhibits polynomial growth in

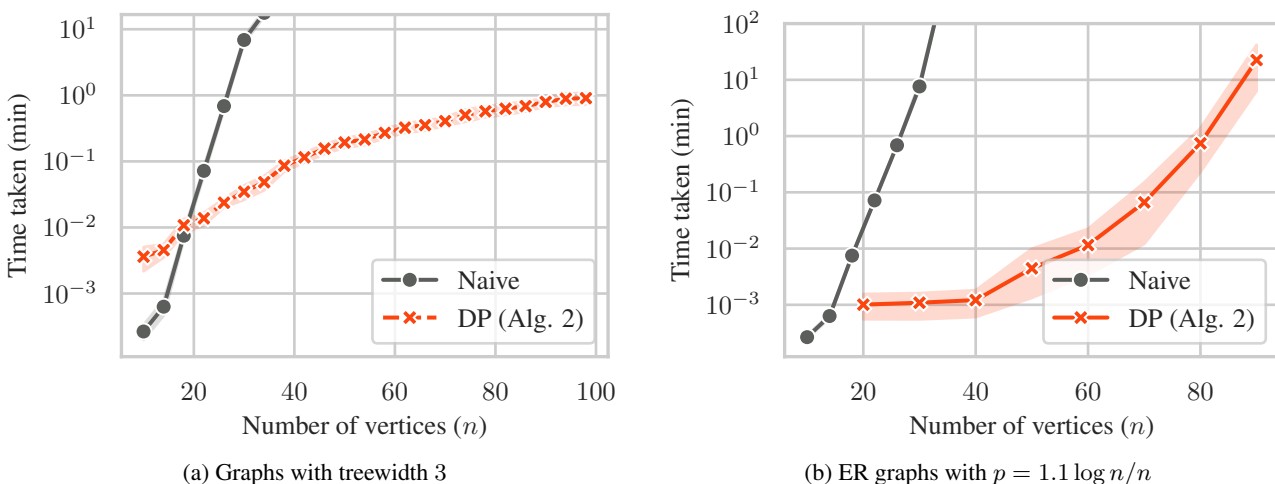

(a) Graphs with treewidth 3

(b) ER graphs with $p = 1.1 \log n/n$

Figure 10: Log running time (in minutes) of naive exhaustive search and our `DP` Algorithm 2 for solving the SRM problem on threewidth-3 and connected ER graphs plotted against the number of vertices, $n$. Shaded regions denote the 99% confidence interval, averaged over 40 runs.

time complexity as $n$ increases, demonstrating that it can handle graphs with $n = 2000$ vertices in about 20 minutes. This is a significant improvement over the exhaustive search algorithm, which is simply infeasible for such large graphs.

### A.9. Empirical Running Time of `Greedy`

Here, we empirically demonstrate that our `Greedy` Algorithm 4 is computationally efficient and can scale to large graphs. We generate connected Erdős-Rényi (ER) graphs with $n \in \{20, \dots, 1000\}$ vertices and edge probability $p = 1.1 \log n/n$. We set the budget $K = 0.2n$ and run both our `DP` and `Greedy` algorithm to solve the SRM problem, measuring their running time. For each $n$, we generate 40 random graphs and report the average running time.

**Results.** Figure 12 shows the average running times of `DP` and `Greedy` across different number of vertices. As shown, `Greedy` is significantly faster than `DP`, with a polynomial growth in time complexity, while `DP` exhibits exponential growth. More importantly, `Greedy` is able to scale to graphs with $n = 1000$ vertices, taking only $\approx 10$ minutes to run.

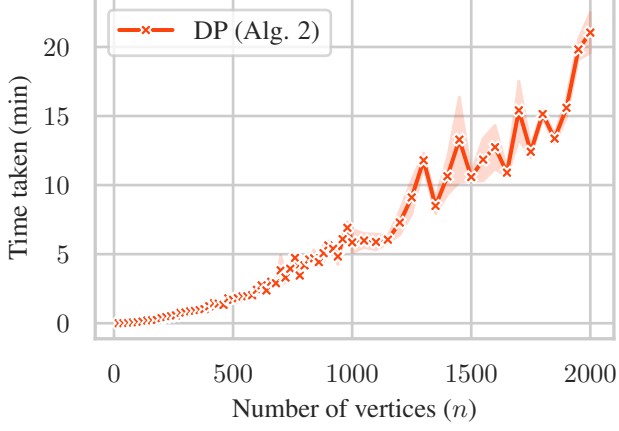
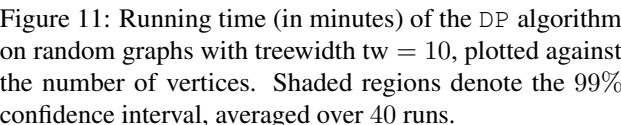

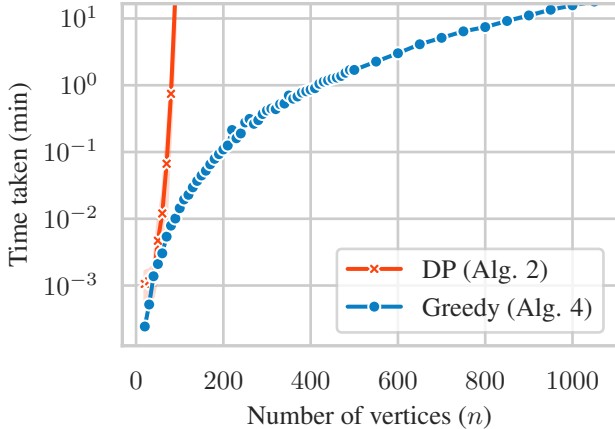

Figure 11: Running time (in minutes) of the `DP` algorithm on random graphs with treewidth tw $= 10$, plotted against the number of vertices. Shaded regions denote the 99% confidence interval, averaged over 40 runs.

Figure 12: Log running time (in minutes) of the `DP` and `Greedy` vaccination algorithms on connected ER graphs, plotted against the number of vertices. Shaded regions denote the 99% confidence interval, averaged over 40 runs.

---

**Algorithm 5** Optimal vaccination for a tree

---

 1: **Input:** Tree $\mathcal{T} = (V, E)$, budget $K$, precision $\varepsilon > 0$
 2: **Output:** Set of vertices to vaccinate $R_\varepsilon$
 3: Initialize $L \leftarrow 0$
 4: Initialize $high \leftarrow \sqrt{|V| - 1}$
 5: Initialize $R_\varepsilon \leftarrow \emptyset$
 6: **while** $high - low > \varepsilon$ **do**
 7: $\quad mid \leftarrow \frac{low + high}{2}$
 8: $\quad$ (feasible, $R$) $\leftarrow$ ALGORITHM 6($\mathcal{T}, K, \lambda = mid$)
 9: $\quad$ **if** feasible **then**
10: $\quad\quad high \leftarrow mid$
11: $\quad\quad R_\varepsilon \leftarrow R$
12: $\quad$ **else**
13: $\quad\quad low \leftarrow mid$
14: $\quad$ **end if**
15: **end while**
16: **Return** $R_\varepsilon$

---

## B. Vaccination on Trees

In this section, we present a vaccination algorithm for trees that we prove to be optimal i.e., it reduces the spectral radius the most for a given vaccination budget $K$. The algorithm is based on the observation that following the removal (i.e., vaccination) of any vertex of a tree, the resulting tree has at least one new connected component.

**Algorithm Overview.** Our algorithm, pseudocode given in Algorithm 5, employs a binary search strategy to identify the smallest possible spectral radius $\lambda_\varepsilon$ achievable by removing at most $K$ vertices from a tree $\mathcal{T}$, up to a precision of $\varepsilon > 0$.

For each candidate value of $\lambda$ during the binary search, we invoke a feasibility check algorithm, pseudocode given in Algorithm 6, to determine whether there exists a set of at most $K$ vertices whose removal ensures that the spectral radius of the resulting forest (i.e., disjoint union of trees) does not exceed $\lambda$. Algorithm 6 operates as follows:

1. **Depth-First Traversal:** The algorithm picks an arbitrary root and recursively explores each subtree in post-order (i.e., children before parent).

2. **Vertex Removal:** If a subtree's spectral radius exceeds $\lambda$, the root of that subtree is marked for removal (vaccination), effectively splitting the tree into smaller components.

3. **Budget Verification:** After the traversal, if the total number of removed vertices does not exceed $K$, the current $\lambda$ is considered feasible.

Using Algorithm 6, we can identify the smallest $\lambda_\varepsilon$ via a binary search procedure conducted within the interval $[0, \sqrt{n-1}]$, leveraging the fact that the spectral radius of a tree is bounded by $\sqrt{n-1}$ (Van Mieghem et al., 2011). The binary search iteratively narrows the range of possible $\lambda$ values by selecting a midpoint and using the feasibility check to determine if the current $\lambda$ is achievable within the vaccination budget. As the domain of $\lambda$ is continuous, we stop the binary search when the interval size is less than a predefined threshold $\varepsilon$.

**Optimality Guarantee.** Below, we formally state and prove the optimality of Algorithm 5.

**Theorem B.1.** *Given a tree $\mathcal{T}$ and vaccination budget $K$, let $\lambda^* = \rho(\mathcal{T}[V \setminus R^*])$, where $R^*$ is the solution to the optimization problem given in Equation (1), and let $\lambda_\varepsilon = \rho(\mathcal{T}[V \setminus R_\varepsilon])$, where $R_\varepsilon$ is the output of Algorithm 5 with precision parameter $\varepsilon$. Then, $|\lambda^* - \lambda_\varepsilon| \leq \varepsilon$.*

*Proof.* Proof follows from the correctness and completeness of Algorithm 6, as detailed in Appendix D.4.4. $\square$

---

**Algorithm 6** Feasibility checker for tree vaccination

---

1: **Input:** Tree $\mathcal{T} = (V, E)$, budget $K$, threshold $\lambda$
2: **Output:** Feasibility and set of vertices to remove $R$
3: Choose an arbitrary root vertex $r \in V$
4: Initialize $R \leftarrow \emptyset$
5: DFS($\mathcal{T}$, $r$, None, $\lambda$, $R$)
6: **if** $|R| \leq k$ **then**
7:     **Return** True, $R$
8: **else**
9:     **Return** False, $R$
10: **end if**

---

1: **function** DFS($\mathcal{T}$, $u$, parent, $\lambda$, $R$)
2:     $S \leftarrow \emptyset$
3:     **for** $v \in \mathcal{N}(u)$ **do**
4:         **if** $v \neq$ parent **then**
5:             $S \leftarrow S \cup \text{DFS}(\mathcal{T}, v, u, \lambda, R)$
6:         **end if**
7:     **end for**
8:     **if** $\rho(\mathcal{T}[S]) > \lambda$ **then**
9:         $R \leftarrow R \cup \{u\}$
10:         **return** $\emptyset$
11:     **else**
12:         **return** $S$
13:     **end if**
14: **end function**

---

**Computational Complexity.** The worst-case complexity of the algorithm is $\mathcal{O}\left(n^3 \log(\sqrt{n}/\varepsilon)\right)$. The complexity arises from (i) the binary search, which has a complexity of $\mathcal{O}(\log(\sqrt{n}/\varepsilon))$; (ii) the post-order traversal of the tree, which has a complexity of $\mathcal{O}(n)$; and (iii) the spectral radius computation, which has a complexity of $\mathcal{O}(n^2)$ using the Lanczos algorithm for sparse graphs with $\mathcal{O}(n)$ nonzero entries (Cullum & Donath, 1974).

## C. Tree Decomposition

In this section, we provide a brief introduction to treewidth, tree decompositions, and nice tree decompositions. For more details, we refer the reader to Cygan et al. (2015).

**Tree decomposition.** A tree decomposition of a graph $\mathcal{G}$ is a tree with bags of vertices as nodes that captures the connectivity and structure of the graph. Below, we provide a formal definition of a tree decomposition.

**Definition C.1.** *(Tree Decomposition from Cygan et al. (2015)). A tree decomposition of $\mathcal{G}$ is a pair $\mathscr{T} = \left(\mathcal{T}, \{X_t\}_{t \in V(\mathcal{T})}\right)$, where $\mathcal{T}$ is a tree whose every node $t$ is assigned a vertex subset $X_t \subseteq V(G)$, called a bag, such that the following three conditions hold:*

- *$\bigcup_{t \in V(\mathcal{T})} X_t = V(\mathcal{G})$. In other words, every vertex of $\mathcal{G}$ is in at least one bag.*

- *For every $u, v \in E(\mathcal{G})$, there exists a node $t$ of $\mathcal{T}$ such that bag $X_t$ contains both $u$ and $v$.*

- *For every $u \in V(\mathcal{G})$, the set $\mathcal{T}_u = \{t \in V(\mathcal{T}) : u \in X_t\}$, i.e., the set of nodes whose corresponding bags contain $u$, induces a connected subtree of $\mathcal{T}$.*

The width of tree decomposition $\mathscr{T}$ equals $\max_{t \in V(\mathcal{T})} |X_t| - 1$, that is, the maximum size of its bag minus 1. Notice that a trivial tree decomposition is formed by a single node with the entire vertex set of the graph as its bag, resulting in a width of $n - 1$.

**Treewidth.** Then, the treewidth of an undirected graph $\mathcal{G}$, denoted by tw($\mathcal{G}$), is the minimum width of any tree decomposition of $\mathcal{G}$. Intuitively, the treewidth of an undirected graph $\mathcal{G}$ is a measure of its tree-likeness (treewidth of a tree is 1, hence the minus 1 in the definition). It is a very important concept used in parametrized algorithms and complexity theory, where many NP-hard problems, such as the Hamiltonian path problem, Steiner tree problem, and many others (Cygan et al., 2022), become tractable on graphs of bounded treewidth.

For a general graph, an upper bound on the treewidth is the number of vertices. However, for many real-world graphs, the treewidth is much smaller than that, making it a useful parameter for designing efficient algorithms.

**Nice tree decomposition.** A nice tree decomposition is a tree decomposition that is easy to write dynamic programming algorithms on. It is formally defined below.

**Definition C.2.** *(Nice tree decomposition from Cygan et al. (2015)). A rooted tree decomposition $\mathscr{T} = \left(\mathcal{T}, \{X_t\}_{t \in V(\mathcal{T})}\right)$ with root $r \in \mathcal{T}$ is nice if $X_r = \emptyset$ and every node of $\mathcal{T}$ is of one of the following four types:*

- **Leaf node**: *a node $t$ that has no children and has $X_t = \emptyset$.*

- **Introduce node**: *a node $t$ with exactly one child $t'$ such that $X_t = X_{t'} \cup \{v\}$ for some vertex $v \notin X_{t'}$; we say that $v$ is introduced at $t$.*

- **Forget node**: *a node $t$ with exactly one child $t'$ such that $X_t = X_{t'} \backslash \{w\}$ for some vertex $w \in X_{t'}$; we say that $w$ is forgotten at $t$.*

- **Join node**: *a node $t$ with two children $t_1, t_2$ such that $X_t = X_{t_1} = X_{t_2}$.*

Working with nice tree decompositions simplifies the design of dynamic programming algorithms, due to limited types of nodes. The following lemma shows that every tree decomposition can be converted to a nice tree decomposition without increasing the width and in polynomial time.

**Lemma C.1.** *(Lemma 7.4 of Cygan et al. (2015)). If a graph $\mathcal{G}$ admits a tree decomposition of width at most $\omega$, then it also admits a nice tree decomposition of width at most $\omega$. Moreover, given a tree decomposition $\left(\mathcal{T}, \{X_t\}_{t \in V(\mathcal{T})}\right)$ of $\mathcal{G}$ of width at most $\omega$, one can in time $\mathcal{O}\left(\omega^2 \cdot \max(|V(\mathcal{T})|, |V(\mathcal{G})|)\right)$ compute a nice tree decomposition of $\mathcal{G}$ of width at most $\omega$ that has at most $\mathcal{O}(\omega|V(\mathcal{G})|)$ nodes.*

## D. Additional Proofs

### D.1. Coupling between Processes $\{Y^{(t)}\}$ and $\{\Psi^{(t)}\}$

The transition probabilites of $\{Y^{(t)}\}_{t \in T_D}$ are governed by the infection and recovery probabilities as described in Section 2. If the chain is in the all zero state, it will remain in that state forever; i.e., $\mathbb{P}(Y^{(t)} = \underline{0} \mid Y^{(t-1)} = \underline{0}) = 1$.

For the modified SIS process $\{\Psi^{(t)}\}_{t \in T_D}$, we define the transition probabilities as follows:

$$\mathbb{P}(\Psi^{(t)} = \psi'' \mid \Psi^{(t-1)} = \psi') = \begin{cases} \mathbb{P}(Y^{(t)} = \psi'' \mid Y^{(t-1)} = \psi'), & \text{if } \psi' \neq \underline{0}, \\ (p_{\text{init}})^{\|\psi''\|}(1 - p_{\text{init}})^{n - \|\psi''\|} & \text{otherwise,} \end{cases}$$

where $\|\psi\| = \sum_{i=1}^{n} \psi_i$ denote the number of 1-elements in a $n$-dimensional vector $\psi \in \{0, 1\}^n$.

Clearly, the processes $\{Y^{(t)}\}_{t \in T_D}$ and $\{\Psi^{(t)}\}_{t \in T_D}$ can be coupled until the first time $t_0$ when $Y^{(t_0)} = \Psi^{(t_0)} = \underline{0}$, at which point process $Y$ goes extinct and process $\Psi$ restarts.

### D.2. Concentration Inequality

Here we state the concentration inequality for the modified SIS Markov chain $\{\Psi^{(t)}\}$ that we use to provide error bounds on the learning algorithm.

**Lemma D.1.** *Let $\{\Psi^{(t)}\}_{t \in T_D}$ be observations from the SIS process, where $T_D$ is a set of (not necessarily consecutive) time indices and $\Psi^{(t)} \in \{0,1\}^V$. Under Assumption 3.1, for any subset $U \subseteq V$, any state $\psi_U \in \{0,1\}^{|U|}$, and any $\varepsilon, \delta > 0$, the following deviation bound holds for the unbiased estimator:*

$$\mathbb{P}\left(\left|\mathbb{P}(\Psi_U = \psi_U) - \hat{\mathbb{P}}(\Psi_U = \psi_U)\right| \geq \varepsilon\right) \leq \delta,$$

*whenever $|T_D| \geq \frac{2\log(2/\delta)}{\varepsilon^2(1-\theta)^2}$, where $0 < \theta < 1$ is a constant depending on the graph structure but independent of $T$.*

*Here, $\mathbb{P}(\Psi_U = \psi_U)$ denotes the true probability (e.g., under the stationary distribution), and $\hat{\mathbb{P}}(\Psi_U = \psi_U) = \varphi_U$ is the empirical estimate over $T = |T_D|$ samples.*

*Proof.* Let $\mathcal{S} = \{0,1\}^V$ be the state space of Markov chain $\{\Psi^{(t)}\}$ with transition probability $p(\psi''|\psi')$ from state $\psi' \in S$ to state $\psi'' \in S$. Let $T = |T_D|$ be the number of observations and let $\mathcal{S}^T$ be the product state space of these $T$ observations. By definition, the process $\{\Psi^{(t)}\}_{t \in \mathbb{N}}$ has the Markov property in time, hence so does a sub-sequence $\{\Psi^{(t)}\}_{t \in T_D}$. We further define $\varphi_U : \mathcal{S}^T \to \mathbb{R}$ as

$$\varphi_U(\Psi^{(1)}, \ldots, \Psi^{(T)}) = \frac{1}{T}\sum_{t=1}^{T} \mathbb{1}\{\Psi_U^{(t)} = \psi_U\}.$$

Notice that the function above is $c$-Lipschitz with respect to the Hamming metric $d(U, W) = \sum_{t=1}^{T} \mathbb{1}\{U^{(t)} \neq W^{(t)}\}$ on $\mathcal{S}^T$, with Lipschitz constant $c = 1/T$.

We are now ready to apply Theorem 1.2 from Kontorovich & Ramanan (2008) for Markov chains. The theorem states that for a $c$-Lipschitz function $\varphi$ on $\mathcal{S}^n$,

$$\mathbb{P}\{|\varphi - \mathbb{E}[\varphi]| \geq t\} \leq 2\exp\left(-\frac{t^2}{2nc^2M_n^2}\right).$$

In our case, the sequence length is $T$ (replacing $n$ in the theorem) and $c = 1/T$. The relevant $M_n$ term becomes $M_T$. The bound becomes

$$\mathbb{P}\{|\varphi_U - \mathbb{E}[\varphi_U]| \geq t\} \leq 2\exp\left(-\frac{t^2}{2T(1/T)^2M_T^2}\right) = 2\exp\left(-\frac{Tt^2}{2M_T^2}\right),$$

where $M_T = (1-\theta^T)/(1-\theta)$ and $\theta$ is the Markov contraction coefficient given by

$$\theta = \sup_{\psi', \psi'' \in \mathcal{S}} \|p(\cdot \mid \psi') - p(\cdot \mid \psi'')\|_{\mathrm{TV}}.$$

We have $\theta < 1$ because every configuration $\psi$ can transfer into the all-zero configuration $\underline{0}$. Hence, for any two states $\psi', \psi'' \in \mathcal{S}$ both $p(\underline{0}|\psi') > 0$ and $p(\underline{0}|\psi'') > 0$ and the support of $p(\cdot|\psi')$ and $p(\cdot|\psi'')$ is not disjoint, which implies $\theta < 1$.

We write $M_T \leq 1/(1-\theta)$ and the probability bound becomes

$$\mathbb{P}\{|\varphi_U - \mathbb{E}[\varphi_U]| \geq t\} \leq 2\exp\left(-\frac{Tt^2(1-\theta)^2}{2}\right).$$

To ensure this probability is at most $\delta$ for a deviation $t = \varepsilon$, we get

$$2\exp\left(-\frac{T\varepsilon^2(1-\theta)^2}{2}\right) \leq \delta$$

Finally, solving for $T$ yields the desired condition

$$T \geq \frac{2\log(2/\delta)}{\varepsilon^2(1-\theta)^2}.$$

$\square$

## D.3. Learning Results

Note that Lemma 3.1, 3.2, and 3.3 hold for both the original process $\{Y^{(t)}\}$ and the modified process $\{\Psi^{(t)}\}$, as they depend only on the transition probabilities outside the all-zero (extinction) state.

**Lemma 3.1.** *If $i, j \in V$ are neighbors, then*

$$\mu_{j|i}^{(t)} \geq p_{inf} \quad \forall t \in [T].$$

*Proof.* Consider an SIS process $\left(Y^{(t)}\right)_{t \in \mathbb{N}}$ at an arbitrary time $t$ with infection probability $p_{\text{inf}}$ and let $i, j \in V$ be neighboring vertices. Let $\|y\| = \sum_{i=1}^{|\mathcal{N}(j)|-1} y_i$ denote the number of 1-elements in a $(|\mathcal{N}(j)| - 1)$-dimensional vector $y \in \{0, 1\}^{|\mathcal{N}(j)|-1}$. Then

$$
\begin{aligned}
\mu_{j|i}^{(t)} &= \mathbb{P}\left(Y_j^{(t+1)} = 1 | Y_j^{(t)} = 0, Y_i^{(t)} = 1\right) \\
&\overset{(a)}{=} \sum_{y \in \{0,1\}^{|\mathcal{N}(j)|-1}} \mathbb{P}\left(Y_j^{(t+1)} = 1 | Y_j^{(t)} = 0, Y_i^{(t)} = 1, Y_{\mathcal{N}(j)\setminus\{i\}}^{(t)} = y\right) \cdot \mathbb{P}\left(Y_{\mathcal{N}(j)\setminus\{i\}}^{(t)} = y | Y_j^{(t)} = 0, Y_i^{(t)} = 1\right) \\
&\overset{(b)}{=} \sum_{y \in \{0,1\}^{|\mathcal{N}(j)|-1}} \left(1 - (1 - p_{\text{inf}})^{\|y\|+1}\right) \cdot \mathbb{P}\left(Y_{\mathcal{N}(j)\setminus\{i\}}^{(t)} = y | Y_j^{(t)} = 0, Y_i^{(t)} = 1\right) \overset{(c)}{\geq} p_{\text{inf}}
\end{aligned}
$$

where $(a)$ is obtained by the law of total probability and $(b)$ by the definition of the SIS process. If the whole neighborhood of $j$ and the state of $j$ is known at time $t$, the transition probability of $Y_j^{(t)}$ from state zero at time $t$ to state one at $t + 1$, is given by $1 - (1 - p_{\text{inf}})^{\|y\|+1}$. That is, we know $Y_j^{(t)} = 0$, $Y_i^{(t)} = 1$ and $Y_{\mathcal{N}(j)\setminus\{i\}}^{(t)} = y$. Hence, the number of infected vertices $\|y\|$ and $i$ contribute to the probability of vertex $j$ being infected in the next round. Finally, $(c)$ is obtained by lower bounding $1 - (1 - p_{\text{inf}})^{\|y\|+1} \geq p_{\text{inf}}$ and by normalization:

$$\sum_{y \in \{0,1\}^{|\mathcal{N}(j)|-1}} \mathbb{P}\left(Y_{\mathcal{N}(j)\setminus\{i\}}^{(t)} = y | Y_j^{(t)} = 0, Y_i^{(t)} = 1\right) = 1.$$

$\square$

**Lemma 3.2.** *Let $j, k \in V$ be non-neighbors and let $S \subseteq V \setminus \{j, k\}$ be a superset of neighbors of $j$, i.e., $\mathcal{N}(j) \subseteq S$. Then*

$$\nu_{j|k,y_S}^{(t)} = 0 \quad \forall t \in [T], \ y_S \in \{0, 1\}^{|S|}.$$

*Proof.* Consider an arbitrary time $t$, vertices $j, k \in V$, $j$ and $k$ non-neighbors, $S \subseteq V \setminus \{j, k\}$ such that $\mathcal{N}(j) \subseteq S$ and a vector $y_S \in \{0, 1\}^{|S|}$. We note that $S$ separates $j$ and $k$, i.e., any path between $j$ and $k$ passes through set $S$. Then by the local Markov property of the model (the probability of a vertex $j$ transitioning from the zero state to the one state only depends on the state of $j$ and its neighbors in the previous round) the evolution of $j$ is conditionally independent of $k$ and we write explicitly:

$$
\begin{aligned}
\nu_{j|k,y_S}^t &= \mathbb{P}\left(Y_j^{(t+1)} = 1 | Y_j^{(t)} = 0, Y_k^{(t)} = 1, Y_S^{(t)} = y_S\right) - \mathbb{P}\left(Y_j^{(t+1)} = 1 | Y_j^{(t)} = 0, Y_k^{(t)} = 0, Y_S^{(t)} = y_S\right) \\
&= \mathbb{P}\left(Y_j^{(t+1)} = 1 | Y_j^{(t)} = 0, Y_{\mathcal{N}(j)}^{(t)} = y_{\mathcal{N}(j)}, Y_k^{(t)} = 1, Y_{S\setminus\mathcal{N}(j)}^{(t)} = y_{S\setminus\mathcal{N}(j)}\right) \\
&\quad - \mathbb{P}\left(Y_j^{(t+1)} = 1 | Y_j^{(t)} = 0, Y_{\mathcal{N}(j)}^{(t)} = y_{\mathcal{N}(j)}, Y_k^{(t)} = 0, Y_{S\setminus\mathcal{N}(j)}^{(t)} = y_{S\setminus\mathcal{N}(j)}\right) \\
&= \mathbb{P}\left(Y_j^{(t+1)} = 1 | Y_j^{(t)} = 0, Y_{\mathcal{N}(j)}^{(t)} = y_{\mathcal{N}(j)}\right) - \mathbb{P}\left(Y_j^{(t+1)} = 1 | Y_j^{(t)} = 0, Y_{\mathcal{N}(j)}^{(t)} = y_{\mathcal{N}(j)}\right) \quad (2) \\
&= 0,
\end{aligned}
$$

where (2) follows from the aforementioned Markov property. $\square$

**Lemma 3.3.** *Let $i, j \in V$ be neighbors and let $S \subseteq V \setminus \{j\}$ be a superset of neighbors of $j$, i.e., $\mathcal{N}(j) \subseteq S$. Then, for any $y_{S\setminus\{i\}} \in \{0, 1\}^{|S|-1}, t \in [T]$*

$$\nu_{j|i,y_{S\setminus\{i\}}}^{(t)} \geq p_{inf}(1 - p_{inf})^{\Delta-1}.$$

*Proof of Lemma 3.3.* Consider an SIS process $\left(Y^{(t)}\right)_{t\in[T]}$ with infection probability $p_{\text{inf}}$ and maximum degree of the underlying graph $\Delta$, let $i, j \in V$ be neighboring vertices, $S \subseteq V \setminus \{j\}$ be a set such that $S \supseteq \mathcal{N}(j)$ and $y_{S\setminus\{i\}} \in \{0,1\}^{|S|-1}$ an arbitrary vector of infection states on set $S \setminus \{i\}$. The conditional influence of $i$ on $j$ given $S$ is defined as:

$$\nu^t_{j|i,y_{S\setminus\{i\}}} = \mathbb{P}\left(Y_j^{(t+1)} = 1 | Y_j^{(t)} = 0, Y_i^{(t)} = 1, Y_{S\setminus\{i\}}^{(t)} = y_{S\setminus\{i\}}\right) - \mathbb{P}\left(Y_j^{(t+1)} = 1 | Y_j^{(t)} = 0, Y_i^{(t)} = 0, Y_{S\setminus\{i\}}^{(t)} = y_{S\setminus\{i\}}\right).$$

As we assume $\mathcal{N}(j) \subseteq S$ and by the local Markov property of the process, $y_{S\setminus\{i\}}$ and the state information on $Y_i^{(t)}$ contain all information for the transition probability of vertex $j$ from the zero state to the one state. Therefore we write:

$$\begin{aligned}
\nu^t_{j|i,y_{S\setminus\{i\}}} &= \mathbb{P}\left(Y_j^{(t+1)} = 1 | Y_j^{(t)} = 0, Y_i^{(t)} = 1, Y_{\mathcal{N}(j)\setminus\{i\}}^{(t)} = y_{\mathcal{N}(j)\setminus\{i\}}\right) \\
&\quad - \mathbb{P}\left(Y_j^{(t+1)} = 1 | Y_j^{(t)} = 0, Y_i^{(t)} = 0, Y_{\mathcal{N}(j)\setminus\{i\}}^{(t)} = y_{\mathcal{N}(j)\setminus\{i\}}\right) \\
&= \left(1 - (1 - p_{\text{inf}})^{\|y_{\mathcal{N}(j)\setminus\{i\}}\|+1}\right) - \left(1 - (1 - p_{\text{inf}})^{\|y_{\mathcal{N}(j)\setminus\{i\}}\|}\right) \\
&= (1 - p_{\text{inf}})^{\|y_{\mathcal{N}(j)\setminus\{i\}}\|}(1 - (1 - p_{\text{inf}})) \\
&\geq p_{\text{inf}}(1 - p_{\text{inf}})^{\Delta-1}.
\end{aligned}$$

where we used the definition of the infection probability and the fact that the degree of a vertex is upper bounded by $\Delta$. $\square$

For the following lemmas, we consider the modified SIS process $\Psi^{(t)}$ described in Appendix D.1 and data collected from its stationary distribution.

**Lemma D.2.** *Under Assumption 3.1, let $i \neq j \in V$ with $m = I(\Psi_j = 0, \Psi_i = 1)$. If $m \geq 2\log(2/\delta)/(\varepsilon^2(1-\theta)^2)$ and $\delta, \varepsilon > 0$, then*

$$\mathbb{P}\left(\left|\mu_{j|i} - \hat{\mu}_{j|i}\right| \geq \varepsilon\right) \leq \delta.$$

*Proof.* Let $i \neq j \in V$, $\delta, \varepsilon > 0$, under Assumption 3.1 let $m = I(\Psi_j = 0, \Psi_i = 1)$ with $m \geq 2\log(2/\delta)/(\varepsilon^2(1-\theta)^2)$. We will use Lemma D.1. First, we set

$$\mathcal{I}(\Psi_j = 0, \Psi_i = 1) = \left\{t \in T_D : \Psi_j^{(t)} = 0, \Psi_i^{(t)} = 1\right\}.$$

We construct a subsequence of the Markov chain $\left\{\Psi^{(t)}\right\}_{t\in[T]}$, which consists of all the time indices directly after an event in $\mathcal{I}$: $\left\{\Psi^{(r+1)} : r \in \mathcal{I}(\Psi_j = 0, \Psi_i = 0)\right\}$. Notice that Lemma D.1 also holds for this Markov chain, since it is itself a Markov chain. Let $S_m = \sum_{r \in \mathcal{I}(\Psi_j=0,\Psi_i=1)} \mathbb{1}\{\Psi_j^{(r+1)} = 1\}$ count the occurrence of $\Psi_j^{(r+1)} = 1$ following a time index satisfying $\Psi_j^{(t)} = 0$ and $\Psi_i^{(t)} = 1$. Then, the expectation of this estimator is given by:

$$\mathbb{E}[S_m] = \mathbb{E}\left[\sum_{r \in \mathcal{I}(\Psi_j=0,\Psi_i=1)} \mathbb{1}\{\Psi_j^{(r+1)} = 1\}\right] \overset{(*)}{=} m\mathbb{P}\left(\Psi_j^{(r+1)} = 1 | \Psi_j^{(t)} = 0, \Psi_i^{(t)} = 1\right),$$

where $(*)$ holds because the expectation is taken over the random variable $\mathbb{1}\{\Psi_j^{(r+1)} = 1\}$ conditioned on the event $\mathcal{I}(\Psi_j = 0, \Psi_i = 1)$ holding in the previous time step. Hence $S_m/m$ can serve as an estimator for $\mathbb{P}\left(\Psi_j^{(r+1)} = 1 | \Psi_j^{(t)} = 0, \Psi_i^{(t)} = 1\right)$. Applying Lemma D.1, we obtain that:

$$\begin{aligned}
&\mathbb{P}\left(\left|\mathbb{P}\left(\Psi_j^{(\cdot+1)} = 1 | \Psi_j^{(\cdot)} = 0, \Psi_i^{(\cdot)} = 1\right) - \hat{\mathbb{P}}\left(\Psi_j^{(\cdot+1)} = 1 | \Psi_j^{(\cdot)} = 0, \Psi_i^{(\cdot)} = 1\right)\right| \geq \varepsilon\right) \\
&= \mathbb{P}\left(\left|\frac{1}{m}\mathbb{E}[S_m] - \frac{1}{m}S_m\right| \geq \varepsilon\right) \leq \delta.
\end{aligned}$$

$\square$

**Lemma D.3.** *Under Assumption 3.1, let $i \neq j \in V$, $S \subseteq V \backslash \{i, j\}$, $\delta, \varepsilon > 0$, $\psi_S \in \{0, 1\}^{|S|}$, $m^+ = I(\Psi_j = 0, \Psi_i = 1, \psi_S)$, and $m^- = I(\Psi_j = 0, \Psi_i = 0, \psi_S)$. If $\min\{m^+, m^-\} \geq \frac{2}{\varepsilon^2(1-\theta)^2} \log(2/\delta)$, then*

$$\mathbb{P}\left(\left|\nu_{j|i,\psi_S} - \hat{\nu}_{j|i,\psi_S}\right| \geq 2\varepsilon\right) \leq 2\delta$$

*Proof.* We will once again use Lemma D.1. Under Assumption 3.1 let $i \neq j \in V$, $S \subseteq V \setminus \{i, j\}$, $\delta, \varepsilon > 0$, $\psi_S \in \{0, 1\}^{|S|}$, $m^+ = I(\Psi_j = 0, \Psi_i = 1, \psi_S)$, $m^- = I(\Psi_j = 0, \Psi_i = 0, \psi_S)$, and $\min\{m^+, m^-\} \geq \frac{2}{\varepsilon^2(1-\theta)^2} \log(2/\delta)$. We define

$$\varphi^+ := \mathbb{P}\left(\Psi_j^{(\cdot+1)} = 1 | \Psi_j^{(\cdot)} = 0, \Psi_i^{(\cdot)} = 1, \Psi_S^{(\cdot)} = \psi_S\right) \quad \text{and} \quad \varphi^- := \mathbb{P}\left(\Psi_j^{(\cdot+1)} = 1 | \Psi_j^{(\cdot)} = 0, \Psi_i^{(\cdot)} = 0, \Psi_S^{(\cdot)} = \psi_S\right).$$

Then, let $\hat{\varphi}^+$ and $\hat{\varphi}^-$ be their estimators, respectively. That is,

$$\hat{\varphi}^+ := \hat{\mathbb{P}}\left(\Psi_j^{(\cdot+1)} = 1 | \Psi_j^{(\cdot)} = 0, \Psi_i^{(\cdot)} = 1, \Psi_S^{(\cdot)} = \psi_S\right),$$
$$\hat{\varphi}^- := \hat{\mathbb{P}}\left(\Psi_j^{(\cdot+1)} = 1 | \Psi_j^{(\cdot)} = 0, \Psi_i^{(\cdot)} = 0, \Psi_S^{(\cdot)} = \psi_S\right).$$

Then, define $S_m^+$ and $S_m^-$ as

$$S_m^+ = \sum_{r \in \mathcal{I}(\Psi_j=0, \Psi_i=1, \psi_S)} \mathbb{1}\{\Psi_j^{(r+1)} = 1\},$$
$$S_m^- = \sum_{r \in \mathcal{I}(\Psi_j=0, \Psi_i=0, \psi_S)} \mathbb{1}\{\Psi_j^{(r+1)} = 1\}.$$

The expectation of the estimators is given by:

$$\mathbb{E}\left[S_m^+\right] = m^+ \varphi^+ \quad \text{and} \quad \mathbb{E}\left[S_m^-\right] = m^- \varphi^-.$$

By application of Lemma D.1:

$$\mathbb{P}\left(\left|\varphi^+ - \hat{\varphi}^+\right| \geq \varepsilon\right) \leq \delta. \tag{3}$$

Similarly,

$$\mathbb{P}\left(\left|\varphi^- - \hat{\varphi}^-\right| \geq \varepsilon\right) \leq \delta. \tag{4}$$

Now, writing $\nu_{j|i,\psi_S}$ and $\hat{\nu}_{j|i,\psi_S}$ using $\varphi^+$ and $\varphi^-$, the triangle inequality implies that

$$\left|\nu_{j|i,\psi_S} - \hat{\nu}_{j|i,\psi_S}\right| = \left|\varphi^+ - \varphi^- - (\hat{\varphi}^+ - \hat{\varphi}^-)\right|$$
$$\leq \left|\varphi^+ - \hat{\varphi}^+\right| + \left|\hat{\varphi}^- - \varphi^-\right|.$$

Combining the above,

$$\mathbb{P}\left(\left|\nu_{j|i,\psi_S} - \hat{\nu}_{j|i,\psi_S}\right| < 2\varepsilon\right)$$
$$\geq \mathbb{P}\left(\left|\varphi^+ - \hat{\varphi}^+\right| + \left|\hat{\varphi}^- - \varphi^-\right| < 2\varepsilon\right)$$
$$\geq \mathbb{P}\left(\{\left|\varphi^+ - \hat{\varphi}^+\right| < \varepsilon\} \cap \{\left|\hat{\varphi}^- - \varphi^-\right| < \varepsilon\}\right) \tag{5}$$
$$\geq 1 - \mathbb{P}\left(\{\left|\varphi^+ - \hat{\varphi}^+\right| \geq \varepsilon\}\right) - \mathbb{P}\left(\{\left|\hat{\varphi}^- - \varphi^-\right| \geq \varepsilon\}\right) \tag{6}$$
$$\geq 1 - 2\delta, \tag{7}$$

where in (5) we have used the intersection of events property, in (6) we have used the union bound, and in (7) we have used (3) and (4).

Finally, taking the complement, we get $\mathbb{P}\left(\left|\nu_{j|i,\psi_S} - \hat{\nu}_{j|i,\psi_S}\right| \geq 2\varepsilon\right) \leq 2\delta$ as desired. $\qquad \square$

Lemmas D.2 and D.3 establish concentration for individual estimators $\hat{\mu}_{j|i}$ and $\hat{\nu}_{j|i,\psi_S}$ (for a specific configuration $\psi_S$). For our learning algorithm SISLearn (Algorithm 1) to succeed, we need these estimates to be accurate simultaneously across all relevant $i, j, S$, and respective configurations $\psi_S$. We thus define the event $\mathcal{A}(\varepsilon_\nu, \varepsilon_\mu)$ that captures this joint accuracy:

$$
\mathcal{A}(\varepsilon_\nu, \varepsilon_\mu) = \left\{ \left( \forall i \neq j \in V : |\mu_{j|i} - \hat{\mu}_{j|i}| \leq \varepsilon_\mu \right) \text{ and} \right.
$$

$$
\left. \left( \forall i \neq j \in V, \forall S \subseteq V \setminus \{i, j\} \text{ s.t. for each respective } \psi_S^* : |\nu_{j|i,\psi_S^*} - \hat{\nu}_{j|i,\psi_S^*}| \leq \varepsilon_\nu \right) \right\},
$$

where, as a reminder, $\psi_S^*(i, j)$, or $\psi_S^*$ when clear from context, is defined as

$$
\psi_S^*(i, j) := \underset{\psi \in \{0,1\}^{|S|-1}}{\arg\max} \min \left( I(\Psi_j^{(t)} = 0, \Psi_i^{(t)} = 1, \Psi_{S\setminus\{i\}}^{(t)} = \psi), I(\Psi_j^{(t)} = 0, \Psi_i^{(t)} = 0, \Psi_{S\setminus\{i\}}^{(t)} = \psi) \right).
$$

The following lemma provides the sample complexity conditions under which this joint event $\mathcal{A}$ holds with given probability.

**Lemma D.4.** *Under Assumption 3.1, if for every $i \neq j \in V$, every $S \subseteq V \setminus \{i, j\}$, and every configuration $\psi_i, \psi_j \in \{0, 1\}$, the number of observations $I(\Psi_i = \psi_i, \Psi_j = \psi_j, \Psi_S = \psi_S^*(i, j))$ is at least $m_{min}$, where*

$$
m_{min} = \frac{2}{(1 - \theta)^2 (\min\{\varepsilon_\nu/2, \varepsilon_\mu\})^2} \log \left( \frac{2n^2(1 + 2^{n-1})}{\zeta} \right),
$$

*then $\mathbb{P}(\mathcal{A}(\varepsilon_\nu, \varepsilon_\mu)) \geq 1 - \zeta$.*

*Proof.* Under Assumption 3.1, assume that for all $i \neq j \in V$, $S \subseteq V \setminus \{i, j\}$, $\psi_i, \psi_j \in \{0, 1\}$, and the respective $\psi_S^*(i, j)$, $I(\Psi_i = \psi_i, \Psi_j = \psi_j, \Psi_S = \psi_S^*(i, j)) \geq \frac{2}{(1-\theta)^2(\min\{\frac{\varepsilon_\nu}{2}, \varepsilon_\mu\})^2} \log\left(\frac{2n^2(1+2^{n-1})}{\zeta}\right)$. Then, we write using a union bound

$$
\mathbb{P}\left(\mathcal{A}(\varepsilon_\nu, \varepsilon_\mu)\right)
$$

$$
= 1 - \mathbb{P}\left(\exists \{i, j\} \subset V, S \subset V \setminus \{i, j\} \text{ such that for } \psi_S^*, |\nu_{j|i,\psi_S^*} - \hat{\nu}_{j|i,\psi_S^*}| > \varepsilon_\nu \text{ or } |\mu_{j|i} - \hat{\mu}_{j|i}| > \varepsilon_\mu\right)
$$

$$
\geq 1 - \mathbb{P}\left( \bigcup_{\substack{i,j \in V \\ S \subseteq V \setminus \{i,j\}}} \left\{ |\nu_{j|i,\psi_S^*} - \hat{\nu}_{j|i,\psi_S^*}| > \varepsilon_\nu \right\} \cup \bigcup_{i,j \in V} \left\{ |\mu_{j|i} - \hat{\mu}_{j|i}| > \varepsilon_\mu \right\} \right)
$$

$$
\geq 1 - \sum_{\substack{i,j \in V \\ S \subseteq V \setminus \{i,j\}}} \mathbb{P}\left( |\nu_{j|i,\psi_S^*} - \hat{\nu}_{j|i,\psi_S^*}| > \varepsilon_\nu \right) - \sum_{i,j \in V} \mathbb{P}\left( |\mu_{j|i} - \hat{\mu}_{j|i}| > \varepsilon_\mu \right).
$$

The number of pairs $(i, j)$ in $V$ can be upper bounded by $n^2$ while the number all conditioning sets $S \subseteq V \setminus \{i, j\}$ is equal to $\sum_{k=1}^{n-2} \binom{n-2}{k} \leq 2^{n-2}$. Let $m = \min_{i,j \in V, S \subseteq V \setminus \{i,j\}} \{I(\Psi_i = \psi_i, \Psi_j = \psi_j, \Psi_S = \psi_S^*)\}$. We obtain therefore from Lemma D.2 and Lemma D.3 that for $m \geq 2\log(2/\delta)/((1-\theta)^2(\min\{\varepsilon_\nu/2, \varepsilon_\mu\})^2)$

$$
\mathbb{P}\left(\mathcal{A}(\varepsilon_\nu, \varepsilon_\mu)\right) \geq 1 - n^2 2^{n-2}(2\delta) - n^2\delta.
$$

Now, since we assumed $m$ satisfies $m \geq \frac{2}{(1-\theta)^2(\min\{\frac{\varepsilon_\nu}{2}, \varepsilon_\mu\})^2} \log\left(\frac{2n^2(1+2^{n-1})}{\zeta}\right)$, we can substitute $\zeta = \delta n^2(1 + 2^{n-1})$, yielding $\mathbb{P}\left(\mathcal{A}(\varepsilon_\nu, \varepsilon_\mu)\right) \geq 1 - \zeta$. $\square$

**Theorem 3.1.** *(Learning Guarantee.)* *Let $\varepsilon_\mu, \varepsilon_\nu, \zeta$ be positive constants, with $\varepsilon_\nu < p_{inf}(1 - p_{inf})^{\Delta - 1}/2$ and $\zeta \leq 1$. Suppose Assumption 3.1 holds. Further, assume that for all $i \neq j \in V$, all $S \subseteq V \setminus \{i, j\}$, and all $\psi_i, \psi_j \in \{0, 1\}$, and the respective $\psi_S^*(i, j)$, the following inequality holds:*

$$
I(\Psi_i = \psi_i, \Psi_j = \psi_j, \Psi_S = \psi_S^*(i, j)) \geq
$$

$$
\frac{2}{(1 - \theta)^2(\min\{\varepsilon_\nu/2, \varepsilon_\mu\})^2} \log\left( \frac{2n^2(1 + 2^{n-1})}{\zeta} \right)
$$

*Then, with probability at least $1 - \zeta$, Algorithm 1, when run with thresholds $\kappa_\nu = \varepsilon_\nu$ and $\kappa_\mu = \varepsilon_\mu$, returns the correct edge set of the underlying graph: $E_\mathcal{D} = E$.*

*Proof.* Assume that for all $i \neq j \in V, S \subseteq V \setminus \{i, j\}, \psi_i, \psi_j \in \{0, 1\}$, and each respective $\psi_S^*(i, j)$ and under Assumption 3.1,

$$I\big(\Psi_i = \psi_i, \ \Psi_j = \psi_j, \ \Psi_S = \psi_S^*\big) \geq \frac{2}{(1 - \theta)^2 (\min\{\varepsilon_\nu/2, \varepsilon_\mu\})^2} \log\left(\frac{2n^2(1 + 2^{n-1})}{\zeta}\right)$$

and $\varepsilon_\nu < p_{\text{inf}}(1 - p_{\text{inf}})^{\Delta-1}/2$. Then, by Lemma D.4, $\mathbb{P}\left(\mathcal{A}(\varepsilon_\nu, \varepsilon_\mu)\right) \geq 1 - \zeta$. For the remainder of the proof we will assume that $\mathcal{A}(\varepsilon_\nu, \varepsilon_\mu)$ holds.

We will show that the algorithm learns the neighborhoods of all vertices correctly one after the other. As the outer for-loop (Line 4–17) iterates over all vertices in $V$, it suffices to show that the algorithm correctly learns the neighborhood of an arbitrary vertex $j \in V$. To do so, we will show that after the inclusion for-loop (Line 6–10), $S$ is a superset of the neighborhood of $j$, and that in the exclusion for-loop (Line 11–15), all neighbors remain in $S$ while all non-neighbors get removed.

**Inclusion Loop.** By Lemma 3.1, if $i$ is a neighbor of $j$, then $\mu_{j|i} \geq p_{\text{inf}}$. Hence, conditioned on $\mathcal{A}$ being true, $\left|\hat{\mu}_{j|i} - \mu_{j|i}\right| \leq \varepsilon_\mu$, and $\hat{\mu}_{j|i} \geq p_{\text{inf}} - \varepsilon_\mu$. Therefore, after the inclusion loop, the set $S$ contains at least all neighbors of $j$, i.e., $S \supseteq \mathcal{N}(j)$.

**Exclusion Loop.** We first establish that no neighboring vertex of $j$ can be removed from $S$ in the exclusion for-loop and next show that all non-neighboring vertices of $j$ are removed from $S$.

Suppose, for the sake of contradiction, that at least one neighboring vertex of $j$ is removed in the exclusion for-loop, say $i$. As shown earlier, $S \supseteq \mathcal{N}(j)$, hence the conditions for Corollary 3.1 are satisfied. As we have also conditioned on $\mathcal{A}(\varepsilon_\nu, \varepsilon_\mu)$, Corollary 3.1, which holds for any $\psi_S$, yields

$$\hat{\nu}_{j|i,\psi_{S\setminus\{i\}}^*} \geq p_{\text{inf}}(1 - p_{\text{inf}})^{\Delta-1} - \varepsilon_\nu.$$

This inequality contradicts the exclusion criterion in Line 12 and thus $i$ cannot be removed from $S$. Hence, no neighboring vertex can be removed by the exclusion loop.

Finally, we know by Lemma 3.2 and $\mathcal{A}(\varepsilon_\nu, \varepsilon_\mu)$ that if $i \notin \mathcal{N}(j)$, then

$$\hat{\nu}_{j|i,\psi_{S\setminus\{i\}}^*} \leq \varepsilon_\nu.$$

We have assumed that $\varepsilon_\nu < p_{\text{inf}}(1 - p_{\text{inf}})^{\Delta-1}/2$. Hence,

$$\hat{\nu}_{j|i,\psi_{S\setminus\{i\}}^*} \leq p_{\text{inf}}(1 - p_{\text{inf}})^{\Delta-1} - \varepsilon_\nu,$$

and so any non-neighboring vertex gets removed.

Thus, after the exclusion for-loop, we have $S = \mathcal{N}(j)$ and the algorithm correctly learns the neighborhood of $j$. Since the outer for-loop iterates over all vertices in $V$, conditioned on $\mathcal{A}$, which occurs with probability at least $1 - \zeta$, the algorithm learns the neighborhood of every vertex in $V$ correctly. $\square$

## D.4. Vaccination Results

### D.4.1. EXTINCTION TIME OF DISCRETE SIS PROCESS

In this section, we state and prove results relating the extinction time of a discrete SIS process to the spectral radius of the underlying graph. This result was first empirically established by Wang et al. (2003), and formally stated and proven for the discrete-time SIS model as defined in Section 2 by Ruhi et al. (2016).

**Lemma D.5.** *Given a discrete SIS model with initial infection probability $p_{init}$, infection probability $p_{inf}$, recovery probability $p_{rec}$, and spreading on graph $\mathcal{G}$ with $n$ vertices and adjacency matrix $\mathbf{A}$, we have that the expected number of infected vertices at time $t$, denoted by $\mathbb{E}[Z^t]$, is upper bounded by $p_{init}\left(1 - p_{rec} + p_{inf}\rho(\mathbf{A})\right)^t n$.*

*Proof.* Denoting the infection probability of vertex $i$ at time $t+1$ by $p_i^{(t+1)} := \mathbb{P}\left(Y_i^{(t+1)} = 1\right)$, we have

$$p_i^{(t+1)} = (1 - p_{\text{rec}}) \cdot p_i^{(t)} + \mathbb{E}_{\{Y_j\}_{j \in \mathcal{N}(i)}}\left[1 - \prod_{j \in \mathcal{N}(i)}\left(1 - p_{\text{inf}} \cdot Y_j^{(t)}\right)\right] \cdot \left(1 - p_i^{(t)}\right), \tag{8}$$

where the first term of the sum is the probability of staying infected, and the second term is the probability of getting infected. We can then bound the product term by using the union bound to get

$$\mathbb{E}\left[1 - \prod_{j \in \mathcal{N}(i)}\left(1 - p_{\text{inf}} \cdot Y_j^{(t)}\right)\right] \leq \mathbb{E}\left[\sum_{j \in \mathcal{N}(i)} p_{\text{inf}} \cdot Y_j^{(t)}\right]$$

$$= \sum_{j \in \mathcal{N}(i)} p_{\text{inf}} \cdot \mathbb{E}\left[Y_j^{(t)}\right]$$

$$= \sum_{j \in \mathcal{N}(i)} p_{\text{inf}} \cdot p_j^{(t)}.$$

Substituting this into Equation (8), we get

$$p_i^{(t+1)} \leq (1 - p_{\text{rec}}) \cdot p_i^{(t)} + p_{\text{inf}} \cdot \sum_{j \in \mathcal{N}(i)} p_j^{(t)} \cdot (1 - p_i^{(t)}).$$

Since $0 \leq (1 - p_i^{(t)}) \leq 1$, we further bound the expression by:

$$p_i^{(t+1)} \leq (1 - p_{\text{rec}}) \cdot p_i^{(t)} + p_{\text{inf}} \cdot \sum_{j \in \mathcal{N}(i)} p_j^{(t)}.$$

Now, let $\mathbf{p}^t = [p_1^t, p_2^t, \ldots, p_n^t]^\top$ represent the vector of infection probabilities at time $t$. The above inequality can be written in matrix form as

$$\mathbf{p}^{t+1} \leq (1 - p_{\text{rec}}) \cdot \mathbf{p}^t + p_{\text{inf}} \cdot \mathbf{A}\mathbf{p}^t$$

$$= \left[(1 - p_{\text{rec}})\mathbf{I} + p_{\text{inf}}\mathbf{A}\right]\mathbf{p}^t$$

$$= \mathbf{M}\mathbf{p}^t,$$

where $I$ is the $n \times n$ identity matrix and $\mathbf{M} = (1 - p_{\text{rec}})I + p_{\text{inf}}\mathbf{A}$ is the transition matrix. Then, by recursively applying the inequality, we obtain

$$\mathbf{p}^t \leq \mathbf{M}^t \mathbf{p}^0,$$

where $\mathbf{p}^0$ is the initial infection probability vector, given by $\mathbf{p}^0 = [p_{\text{init}}, p_{\text{init}}, \ldots, p_{\text{init}}]^\top$. The spectral radius of matrix $\mathbf{M}$, denoted by $\rho(\mathbf{M})$, is given by:

$$\rho(\mathbf{M}) = (1 - p_{\text{rec}}) + p_{\text{inf}}\rho(A),$$

where $\rho(\mathbf{A})$ is the spectral radius of the adjacency matrix $\mathbf{A}$. Now, we can write the expected number of infected vertices at time $t$ as

$$\mathbb{E}[Z^t] = \sum_{i=1}^{n} \mathbb{E}[Y_i^t]$$

$$= \sum_{i=1}^{n} p_i^{(t)}$$

$$= \|\mathbf{p}^t\|_1$$

$$\leq \sqrt{n}\|\mathbf{p}^t\|_2,$$

where the last inequality follows from the Cauchy-Schwarz inequality. Substituting the bound on $\mathbf{p}^t$ into the above expression, we obtain

$$\mathbb{E}[Z^t] \leq \sqrt{n}\|\mathbf{M}^t\mathbf{p}^0\|_2$$
$$\leq \sqrt{n}\rho(\mathbf{M})^t\|\mathbf{p}^0\|_2,$$

where the last inequality follows from $M$ being a symmetric matrix. Simplifying further and substituting in the value of $\rho(\mathbf{M})$, we obtain $\mathbb{E}[Z^t] \leq (1 - p_{\text{rec}} + p_{\text{inf}}\rho(\mathbf{A}))^t np_{\text{init}}$, as desired. $\qquad\square$

**Theorem 4.1.** *The expected extinction time $\mathbb{E}[\tau]$ is upper bounded by $\mathcal{O}(\log n)$ if $\rho(\mathcal{G}) < p_{rec}/p_{inf}$.*

*Proof.* We first define $t' := \arg\inf_t\{\mathbb{E}[Z^t] \leq 1/n\}$ as the first round $t$ such that $\mathbb{E}[Z^t] \leq 1/n$. Then, using the upper bound from Lemma D.5, and assuming that $\rho(\mathbf{A}) < p_{\text{rec}}/p_{\text{inf}}$, we have that if $t' = \lceil -\log(p_{\text{init}}n^2)/\log(\rho(\mathbf{M})) \rceil$, then $\mathbb{E}[Z^{t'}] \leq \rho(\mathbf{M})^{t'}np_{\text{init}} \leq 1/n$. Then, we can write the expected extinction time as

$$\mathbb{E}[\tau] = \mathbb{E}\left[\sum_{t=1}^{\infty} \mathbb{1}\left\{Z^t \geq 1\right\}\right]$$

$$= \sum_{t=1}^{\infty} \mathbb{P}\left(Z^t \geq 1\right)$$

$$= \sum_{t=1}^{t'} \mathbb{P}\left(Z^t \geq 1\right) + \sum_{t=t'+1}^{\infty} \mathbb{P}\left(Z^t \geq 1\right)$$

$$\leq t' + \sum_{t=t'+1}^{\infty} \mathbb{P}\left(Z^t \geq 1\right).$$

Now, we will show that the second term in the above expression goes to zero asymptotically and we will be done. To do so, we will apply Markov's inequality and Lemma D.5 to obtain

$$\sum_{t=t'+1}^{\infty} \mathbb{P}\left(Z^t \geq 1\right) \leq \sum_{t=t'+1}^{\infty} \mathbb{E}[Z^t]$$

$$\leq \sum_{t=t'+1}^{\infty} p_{\text{init}}\rho(\mathbf{M})^t n.$$

Using the definition of $t'$, we have that

$$\sum_{t=t'+1}^{\infty} \mathbb{P}\left(Z^t \geq 1\right) \leq \sum_{k=1}^{\infty} p_{\text{init}}\rho(\mathbf{M})^{t'+k} n$$

$$= \sum_{k=1}^{\infty} \left(p_{\text{init}}\rho(\mathbf{M})^{t'} n\right) \rho(\mathbf{M})^k$$

$$\leq \frac{1}{n}\sum_{k=1}^{\infty} \rho(\mathbf{M})^k$$

$$= \frac{1}{n}\frac{\rho(\mathbf{M})}{1 - \rho(\mathbf{M})},$$

where in the last line we have used the fact that $|\rho(\mathbf{M})| < 1$. Finally, we have that the expected extinction time is upper bounded by

$$\mathbb{E}[\tau] \leq \left\lceil -\frac{\log(p_{\text{init}}n^2)}{\log(\rho(\mathbf{M}))} \right\rceil + \frac{1}{n}\frac{\rho(\mathbf{M})}{1 - \rho(\mathbf{M})}$$
$$\backsim \mathcal{O}(\log n).$$

$\qquad\square$

D.4.2. OPTIMALITY OF ALGORITHM 2

Here, we state and prove the optimality of Algorithm 2.

**Theorem D.1.** *Given a graph $\mathcal{G}$ and vaccination budget $K$, let $\lambda^* = \rho(\mathcal{G}[V \setminus R^*])$, where $R^*$ is the solution to the optimization problem given in Equation* (1), *and let $\lambda_\varepsilon = \rho(\mathcal{G}[V \setminus R_\varepsilon])$, where $R_\varepsilon$ is the output of Algorithm 2 with precision parameter $\varepsilon$. Then, $|\lambda^* - \lambda_\varepsilon| \leq \varepsilon$.*

*Proof.* Proof follows from the correctness and completeness of Algorithm 3, stated and proven below. $\square$

Throughout the rest of the subsection, let $\mathcal{G}$ be an undirected graph with nice tree decomposition $\mathscr{T} = \left(\mathcal{T}, \{X_t\}_{t \in V(\mathcal{T})}\right)$, $K \in \mathbb{Z}^+$ the vaccination budget, and $\lambda \in \mathbb{R}^+$ the desired spectral radius threshold. Assume that Algorithm 3 is run with input $\mathcal{G}$, $\mathscr{T}$, $K$, and $\lambda$.

**Claim D.1** (Correctness of Algorithm 3). *If Algorithm 3 returns* True *and identifies a set $R$ of vertices to vaccinate, then $|R| \leq K$ and $\rho(\mathcal{G}[V \setminus R]) \leq \lambda$.*

*Proof.* Assume that Algorithm 3 returns True with a set $R$. We will prove the correctness of the algorithm by proving the correctness of $\mathrm{DP}[t, S, c]$ for each node type using induction i.e., we will prove that if $\mathrm{DP}[t, S, c] \neq \emptyset$, then all $R \in \mathrm{DP}[t, S, c]$ satisfy simultaneously (i) $|R| = c$, (ii) $S \cap R = \emptyset$ and (iii) $\rho(\mathcal{G}[V_t \setminus R]) \leq \lambda$.

For the base case, notice that $\mathrm{DP}[t, S, c] = \emptyset$ for all $t \in V(\mathcal{T})$, $S \subseteq X_t$, and $0 \leq c \leq K$, so the hypothesis is trivially satisfied. Then, for the induction step, notice that the algorithm iterates through the tree's nodes in a post-order and visits children before the parent. Thus, for each node $t$, we assume the correctness of its children's DP values (induction hypothesis). We will now prove the correctness of the node $t$, based on its type.

**Leaf node.** For a leaf node $t$, note that $V_t = \emptyset$, and the algorithm sets the only $R$ in $\mathrm{DP}[t, \emptyset, 0]$ as $\emptyset$, which trivially satisfies (i), (ii), and (iii).

**Introduce node.** For an introduce node $t$ with child node $t'$, we have $X_t = X_{t'} \cup \{v\}$ for some $v \in V(\mathcal{G})$. For all $S \subseteq X_t$ and $0 \leq c \leq K$,

- If $v \notin S$, then the algorithm sets $\mathrm{DP}[t, S, c] = \{R' \cup \{v\} : R' \in \mathrm{DP}[t', S, c - 1]\}$. Thus, for any $R' \cup \{v\} = R \in \mathrm{DP}[t, S, c]$, we have $|R| = |R'| + 1 = c$ and $S \cap R = (S \cap R') \cup (S \cap \{v\}) = \emptyset$, so (i) and (ii) are satisfied. For (iii), notice that $V_t \setminus \{v\} = V_{t'}$, so $\rho(\mathcal{G}[V_t \setminus R]) = \rho(\mathcal{G}[V_{t'} \setminus R']) \leq \lambda$ by the induction hypothesis.

- If $v \in S$, then the algorithm sets $\mathrm{DP}[t, S, c] = \{R' \in \mathrm{DP}[t', S \setminus \{v\}, c] : \rho(G[V_t \setminus R']) \leq \lambda\}$. Therefore, (iii) is satisfied by definition, and (i) and (ii) are satisfied by the induction hypothesis ($|R'| = c$ and $S \cap R' = \emptyset$).

**Forget node.** For a forget node $t$ with child node $t'$, we have $X_t = X_{t'} \setminus \{w\}$ for some $w \in V(\mathcal{G})$. Thus, $V_t = V_{t'}$. For all $S \subseteq X_t$ and $0 \leq c \leq K$, the algorithm sets $\mathrm{DP}[t, S, c] = \mathrm{DP}[t', S, c] \cup \mathrm{DP}[t', S \cup \{w\}, c]$. Let $R \in \mathrm{DP}[t, S, c]$.

- If $R \in \mathrm{DP}[t', S, c]$, then $|R| = c$, $S \cap R = \emptyset$, and $\rho(\mathcal{G}[V_t \setminus R]) = \rho(\mathcal{G}[V_{t'} \setminus R]) \leq \lambda$ by the induction hypothesis.

- If $R \in \mathrm{DP}[t', S \cup \{w\}, c]$, then $|R| = c$ and $\rho(\mathcal{G}[V_t \setminus R]) \leq \lambda$ by the induction hypothesis, so (i) and (iii) are satisfied. For (ii), notice that $(S \cup \{w\}) \cap R = \emptyset$, by the induction hypothesis, implying $S \cap R = \emptyset$.

**Join node.** For a join node $t$ with child nodes $t_1$ and $t_2$, we have $X_t = X_{t_1} = X_{t_2}$. For all $S \subseteq X_t$ and $0 \leq c \leq K$, the algorithm sets

$$\mathrm{DP}[t, S, c] =$$
$$\left\{ R_1 \cup R_2 \,\middle|\, 0 \leq c_1, c_2 \leq K, \; R_1 \in \mathrm{DP}[t_1, S, c_1], \; R_2 \in \mathrm{DP}[t_2, S, c_2], \; |R_1 \cup R_2| = c, \; \rho(G[V_t \setminus (R_1 \cup R_2)]) \leq \lambda \right\}.$$

Conditions (i) and (iii) are satisfied by definition. Condition (ii) is also satisfied as $S \cap R_1 = S \cap R_2 = \emptyset$ by the induction hypothesis.

We have shown that assuming the correctness a node $t$'s children's DP values, the correctness of $t$'s DP values follows. This, combined with the base case, establishes the correctness of the algorithm through induction. $\qquad\square$

**Claim D.2** (Completeness of Algorithm 3)**.** *If there exists a set $R' \subseteq V(\mathcal{G})$ with $|R'| \leq K$ such that $\rho(\mathcal{G}[V(\mathcal{G}) \setminus R']) \leq \lambda$, then Algorithm 3 will return* `True` *along with a set $R \subseteq V$ with $|R| \leq K$ such that $\rho(\mathcal{G}[V(\mathcal{G}) \setminus R]) \leq \lambda$.*

*Proof.* Let $R \subseteq V(\mathcal{G})$ be any set of vertices with $|R| \leq K$ for which $\rho\big(\mathcal{G}[V(\mathcal{G}) \setminus R]\big) \leq \lambda$. We show by (bottom-up) induction on the nodes of the tree decomposition $\mathcal{T}$ that $R$ is recorded in the table $\mathrm{DP}[\cdot, \cdot, \cdot]$. In particular, we will prove that for every node $t \in V(\mathcal{T})$,

$$R_t := R \cap V_t \quad \text{belongs to} \quad \mathrm{DP}\big[t, \ X_t \cap (V(\mathcal{G}) \setminus R), \ |R_t|\big].$$

Since $X_t \cap R = \emptyset$ is equivalent to $X_t \cap (V(\mathcal{G}) \setminus R) = X_t$, we define the 'preserved set' $S_t := X_t \setminus R$ so that $R_t$ never intersects $S_t$. As before, $V_t$ denotes the union of all bags in the subtree of $\mathcal{T}$ rooted at $t$.

**Base Case (Leaf node).** If $t$ is a leaf node, then $X_t = \emptyset$ and $V_t = \emptyset$. Thus $R_t = R \cap \emptyset = \emptyset$. The algorithm sets $\mathrm{DP}[t, \emptyset, 0] = \{\emptyset\}$ (and $\mathrm{DP}[t, S, c] = \emptyset$ otherwise). Clearly, $R_t = \emptyset$ is indeed recorded in $\mathrm{DP}[t, \emptyset, 0]$.

**Induction Hypothesis.** Assume that for every child $t'$ of $t$, $R_{t'} \in \mathrm{DP}[t', S_{t'}, |R_{t'}|]$ where $S_{t'} = X_{t'} \setminus R$. We now prove that $R_t \in \mathrm{DP}[t, S_t, |R_t|]$.

**Introduce node.** Suppose $t$ is an introduce node with child $t'$, and let $v$ be the vertex introduced at $t$, so $X_t = X_{t'} \cup \{v\}$. There are two cases:

- **Case 1:** $v \in R$. Then $R_t = R_{t'} \cup \{v\}$ and $S_t = S_{t'}$ (since $v \notin S_t$). By induction, we know $R_{t'} \in \mathrm{DP}[t', S_{t'}, |R_{t'}|]$. Because $v \in R_t$, the DP update rule places $R_t$ into $\mathrm{DP}[t, S_t, |R_t|]$ as $R_{t'} \cup \{v\}$.

- **Case 2:** $v \notin R$. Then $R_t = R_{t'}$ and $S_t = S_{t'} \cup \{v\}$. By induction, $R_{t'} \in \mathrm{DP}[t', S_{t'} \cup \{v\}, |R_{t'}|]$. The DP update rule copies $R_{t'}$ directly into $\mathrm{DP}[t, S_t, |R_t|]$ (and also checks $\rho(\mathcal{G}[V_t \setminus R_t]) \leq \lambda$, which holds by assumption).

**Forget node.** Suppose $t$ is a forget node with child $t'$, and let $w$ be the vertex forgotten at $t$, so $X_t = X_{t'} \setminus \{w\}$ and $V_t = V_{t'}$. We again have two cases:

- **Case 1:** $w \in R$. Then $R_t = R_{t'}$ and $S_t = S_{t'} \setminus \{w\}$. By induction, $R_{t'} \in \mathrm{DP}[t', S_{t'}, |R_{t'}|]$. The DP update rule ensures $R_{t'}$ appears in $\mathrm{DP}[t, S_t, |R_t|]$.

- **Case 2:** $w \notin R$. Then $R_t = R_{t'}$ and $S_t = S_{t'}$. By induction, $R_{t'} \in \mathrm{DP}[t', S_{t'}, |R_{t'}|]$. Again, $R_{t'}$ is copied into $\mathrm{DP}[t, S_t, |R_t|]$.

**Join node.** Suppose $t$ is a join node with two children $t_1$ and $t_2$, where $X_t = X_{t_1} = X_{t_2}$. Observe that $R_t = R_{t_1} \cup R_{t_2}$ and $S_t = S_{t_1} = S_{t_2}$. By the induction hypothesis,

$$R_{t_1} \in \mathrm{DP}[t_1, S_{t_1}, |R_{t_1}|] \quad \text{and} \quad R_{t_2} \in \mathrm{DP}[t_2, S_{t_2}, |R_{t_2}|].$$

The join node's DP rule unites these sets whenever they align on the same $S_t$ and total size $|R_t| = |R_{t_1}| + |R_{t_2}| \leq K$. By our feasibility assumption, $\rho(\mathcal{G}[V_t \setminus (R_{t_1} \cup R_{t_2})]) \leq \lambda$, so $R_t$ is placed in $\mathrm{DP}[t, S_t, |R_t|]$.

We have shown that whenever the children of a node $t$ have valid DP entries, the node $t$ will also have a valid DP entry. Thus, by induction, the solution $R'$ is recorded in the root node's DP table, and the algorithm returns `True`. More specifically, at the root node $r$ we have $R \in \mathrm{DP}[r, \emptyset, c]$ for some $0 \leq c \leq K$. Therefore, $\mathrm{DP}[r, \emptyset, c] \neq \emptyset$, which guarantees Algorithm 3 detects a feasible solution. $\qquad\square$

### D.4.3. Time Complexity of Algorithm 2

**Theorem 4.2.** *Given an input graph $\mathcal{G}$ with treewidth $\mathrm{tw}(\mathcal{G}) \leq \omega$, budget $K$, and precision $\varepsilon$, Algorithm 2 has a worst-case time complexity of $\mathcal{O}\big(n^{\mathcal{O}(1)} K^{\mathcal{O}(1)} 2^{\mathcal{O}(\omega)} \log(\Delta/\varepsilon)\big)$.*

*Proof.* The complexity arises from the binary search, building the DP table, node and spectral radius computation, each of which we analyze below.

**Binary search.** The binary search is done on $[0, \Delta]$ and runs until $\text{high} - \text{low} > \varepsilon$, thus taking $\mathcal{O}(\log(\Delta/\varepsilon))$ time.

**DP table.** The DP table is indexed by $t \in V(\mathcal{T})$, $S \subseteq X_t$, and $c \in \{0, 1, \dots, K\}$. The number of nodes in the tree decomposition is upper bounded by $\mathcal{O}(\omega n)$ by Lemma C.1. Then, as size of each bag $X_t$ is at most $\omega + 1$ by the definition of treewidth (see Appendix C), we have that the total number of subsets is upper bounded by $\mathcal{O}(2^{\omega+1})$. Thus, the total number of entries in the DP table is $\mathcal{O}(\omega n 2^{\omega+1} K)$.

Note that although the DP table is described in Section 4.2 as storing all feasible solutions per $(t, S, c)$ triplet, this is not necessary. We only need to retain a representative solution for each state, and in particular only before a join node is reached in the post-order. In other words, once a join node has been processed, it suffices to only keep one solution per triplet. This is how we avoid a $\binom{n}{K}$ factor in the complexity. We still need to perform a search over $K^2$ many $c$ values at each node join node, but this only adds a $\mathcal{O}(K^2)$ factor to the complexity.

**Spectral radius.** The spectral radius computation complexity is at most $\mathcal{O}(n^2)$ using the Lanczos algorithm for sparse graphs with $\mathcal{O}(n)$ nonzero entries (Cullum & Donath, 1974).

Combining the above, we have that the total time complexity is $\mathcal{O}\left(n^3 K^3 \omega 2^{\omega+1} \log(\Delta/\varepsilon)\right)$. $\qquad\square$

### D.4.4. OPTIMALITY OF ALGORITHM 5

Here, we prove the optimality of Algorithm 5, which follows directly from the correctness and completeness of Algorithm 6, proven below.

Throughout the rest of this subsection, let $\mathcal{T} = (V, E)$ be an undirected tree, $K \in \mathbb{Z}^+$ the vaccination budget, and $\lambda \in \mathbb{R}^+$ the desired spectral radius threshold. Assume that Algorithm 6 is run with input $\mathcal{T}'$, $K$, and $\lambda$. Without loss of generality, order the vertices in the tree $\mathcal{T}$ by picking a random root and ordering the other vertices accordingly.

Lastly, define $\mathcal{C}(\mathcal{T})$ as the set of connected components of $\mathcal{T}$, where $\mathcal{T}$ is a forest, and define a *minimal offending subtree* as a connected subtree $S \subseteq V$ where $\rho(\mathcal{T}[S]) > \lambda$ and for any proper connected subtree $S' \subset S$, $\rho(\mathcal{T}[S']) \leq \lambda$.

**Claim D.3** (Correctness). *If Algorithm 6 returns* `True` *and identifies a set $R$ of vertices to remove, then $|R| \leq k$ and after removing $R$ from the tree $\mathcal{T}$, $\rho(\mathcal{T}[V \setminus R]) \leq \lambda$.*

*Proof.* Assume that Algorithm 6 returns `True` with a set $R$. First, observe that $|R| \leq k$, due to the if clause in line 6. Then, notice that the algorithm performs a post-order depth-first traversal of the tree. At each vertex $u$, it attempts to merge all child subtrees with $u$.

If the spectral radius $\rho(\mathcal{T}[S]) \leq \lambda$, the subtree remains connected, and no cut is made. Otherwise, the algorithm adds $u$ to $R$, effectively removing $u$ from its parent by not including $u$ in the merged subtree, ensuring that the offending subtree rooted at $u$ does not propagate upwards.

Since the traversal is post-order, all descendants of any vertex $u$ have been processed before $u$ itself. Therefore, when an offending subtree is detected at $u$, none of its descendants have subtrees with spectral radius larger than $\lambda$. Consequently, when the final vertex in the post-order is reached, all subtrees rooted at all other vertices satisfy the spectral radius threshold.

Thus, after Algorithm 6 terminates, we have that for all connected components (i.e., subtrees) after removal of vertices in $R$, $S \in \mathcal{C}(\mathcal{T}[V \setminus R])$, $\rho(\mathcal{T}[S]) \leq \lambda$. Then, the desired claim follows from applying the equality relating the spectral radius of a graph to the maximum of the spectral radii of its connected components: $\rho(\mathcal{T}[V \setminus R]) = \max_{S \in \mathcal{C}(\mathcal{T}[V \setminus R])} \rho(\mathcal{T}[S])$ (Van Mieghem et al., 2011). $\qquad\square$

**Claim D.4** (Completeness). *If there exists a set $R' \subseteq V$ with $|R'| \leq K$ such that $\rho(\mathcal{T}[V \setminus R']) \leq \lambda$, then Algorithm 6 will return* `True` *along with a set $R \subseteq V$ with $|R| \leq K$ such that $\rho(\mathcal{T}[V \setminus R]) \leq \lambda$.*

*Proof.* Assume, for contradiction, that there exists a feasible solution $R'$ with $|R'| \leq K$ such that $\rho(\mathcal{T}[V \setminus R']) \leq \lambda$, but Algorithm 6 returns `False`, implying $|R| > K$, where $R$ is the set constructed by Algorithm 6.

When the algorithm processes the tree in post-order, it identifies and removes the root of each minimal offending subtree. Moreover, following each removal of a subtree root, that subtree is never visited again by the algorithm. Thus, each element of $R$ corresponds to removing exactly one vertex from a distinct minimal offending subtree.

Now if $|R| > K$, this implies that there are more than $K$ distinct minimal offending subtrees in $\mathcal{T}$. However, since each such subtree requires at least one vertex removal and $|R'| \leq K$, it is impossible to cover all offending subtrees with $K$ removals. This is a contradiction and thus Algorithm 6 must return `True`, establishing the optimality of the algorithm. $\square$

