# OpenReview forum: "Learn to Vaccinate: Combining Structure Learning and Effective Vaccination for Epidemic and Outbreak Control"
_ICML.cc/2025/Conference — ICML 2025 poster_

### Official Review · Reviewer_kFRe · 2025-03-13

**Overall Recommendation:** 2

**Summary:**

The paper studies an important problem of picking nodes to vaccinate in a network,
assuming an SIS model of epidemic spread. The authors assume the network is not known,
and needs to be learned. This is an interesting extension, since most prior work assumes
the network is known. The authors present an algorithm for optimal vaccination set
when the network has bounded treewidth. In their experiments, they use another algorithm
which greedily reduces the spectral radius the most in each step. The authors show
their algorithms have slightly better performance compared to many other baselines.

**Claims And Evidence:**

Seem ok

**Essential References Not Discussed:**

Some others on network inference, e.g., (Abrahao et al., Trace Complexity of Network Inference,
2013), and for characterization of the SIS model, e.g., (Ganesh et al., INFOCOM 2005)

**Experimental Designs Or Analyses:**

Seem ok. Prior work on network construction using SIR model gives bounds on
the sample complexity. The authors mention Theorem 3.1 bounds the sample complexity.
That is not clear, and it is not clear how this is done in the experiments.

**Methods And Evaluation Criteria:**

While it is ok to assume a meta-stable state, as in (Van De Bovenkamp & Van Mieghem, 2014),
I am not sure it justifies the stationary distribution of the form stated in section 3.2,
from which the authors are making assumption 3.1. This needs more discussion, and the response doesn't seem adequate. The graphs learned can have large treewidth, and the greedy algorithm is needed to ensure poly time

**Other Comments Or Suggestions:**

Please see above

**Other Strengths And Weaknesses:**

The setup is interesting, since networks are not fully known. The results are promising.

However, the seems limiting that the results are shown for very specific parameters.
So not sure how this works for the broader parameter regime.

The DP algorithm only works for treewidth bounded networks. What is the bound on the
treewidth of the inferred networks? The authors would need another step for finding the
treewidth, which can only be approximated.

How data is used in the network inference step is not very clear, unlike the other
network inference problems. Do you need to keep observing the infection states for
a long time?

Finally, as mentioned above, assuming infection states are fully observed but not the edges doesn't seem any more realistic or practical, and needs better motivation

**Questions For Authors:**

Please address the weaknesses mentioned above:
-- results are shown for very specific parameters, performance in other regimes
-- treewidth of inferred network. Can it be large? If so, how will the DP algorithm be used?
-- how long do you need to observe, sample complexity
-- model of observation of infection states

**Relation To Broader Scientific Literature:**

Prior work on vaccination strategies has assumed the network is known. So the setup here
is interesting, though it is not clear how practical this is. It is assumed that all
infections are detected, but edges are not known. In practice (as during COVID), infection
states seem just as sensitive (or maybe more). So maybe a more realistic combination of
observations of edges and infection states is realistic

**Theoretical Claims:**

Seem ok, but haven't verified all the proofs in the appendix

---

> ### Author Rebuttal · Authors · 2025-03-31
>
> We thank the reviewer for their time and thoughtful comments. We address each of the points raised below and will clarify them in the revised manuscript.
>
> ### Stationarity
> We refer the reviewer to our response to Q6 of Reviewer Gdj4 for a more detailed response. In short, either the process dies out quickly, in which case vaccination is unnecessary, or it enters a meta-stable regime where it behaves like a Markov chain with a stationary distribution, justifying Assumption 3.1.
>
> ### Comparison with the SIR model and sample complexity
> We thank the reviewer for this observation. Prior work on SIR-based structure learning relies on observing multiple independent cascades, such as Gomez-Rodriguez et al. 2012, since in SIR models each node can be infected at most once. As a result, interactions between nodes are limited within each cascade, and restarting the process is necessary to collect sufficient signal.
>
> In contrast, the SIS model permits reinfection, enabling a single persistent cascade to generate rich temporal correlations over time. Our learning algorithm leverages this property and thus does not require multiple independent cascades. Consequently, our sample complexity guarantees given in Theorem 3.1 are expressed in terms of the number of times specific infection patterns (e.g., $I(Y_j = 0, Y_i = 1, Y_S = y_S)$) appear in the data, rather than the number of cascades.
>
> Empirically, as shown in Figure 3 (Appendix A.2), SISLearn achieves an F1 score of 0.80 with just 400 rounds of observation, demonstrating strong learning performance.
>
> ### Observation of edges vs infection states
> We agree that combining partial edge and infection information is a realistic and important direction. We note that the setting of unknown edges but observed infection states is standard in prior work on SIR-based structure learning, like Gomez-Rodriguez et al., 2012 or Netrapalli & Sanghavi, 2012, which motivates exploring the analogous setup in the SIS setting.
>
> That said, our approach is highly flexible. Since SISLearn learns the neighborhood of each vertex independently, it can easily incorporate partial edge knowledge. Similarly, it can tolerate missing infection data: as long as some infection history is observed for every vertex and pairs of consecutive observations (e.g., at times $t$ and $\(t{+}1\)$) are available, the algorithm remains effective, though naturally requiring more samples.
>
> ### Specific parameters
> We emphasize that our parameter choices were guided by the goal of evaluating non-trivial regimes of the SIS model—where vaccinations are both necessary and impactful. If the infection-to-recovery ratio is too low, the disease naturally dies out without intervention, making vaccination redundant. If it is too high, the process becomes effectively unstoppable without unrealistic levels of vaccination. The intermediate regime is where extinction dynamics are sensitive to targeted interventions—precisely the setting where intelligent strategies matter.
>
> To demonstrate robustness beyond this regime, we conducted additional experiments on both low (Figure 5) and high (Figure 6) infection-to-recovery ratios; results are provided [here](https://drive.proton.me/urls/5N1PC0M6ZW#oCweKFODmBNn). As can be observed, our combined learning-and-vaccination approach still outperforms the other baselines. If the reviewer has a specific parameter regime in mind, we would be glad to run additional experiments.
>
> ### Tree width of inferred networks
> The China flu graphs (Sec. 5) and USA flu graphs (App. A) have treewidths of 8 and 58, respectively, computed exactly using the state-of-the-art treewidth solver from the PACE 2017 competition (Tamaki, 2019). For graphs with large treewidth (tw > 20), our proposed Greedy algorithm is preferable due to its polynomial-in-n complexity, making it significantly more efficient. Please also see our answer to the computational feasibility comment of Reviewer WHtL, where we discuss new experiments (new Figures 1 & 2).
>
> ### Usage of data & observation horizon
> Unlike SIR-based inference methods that rely on multiple independent cascades, SISLearn uses infection data from a single, ongoing epidemic, needing only the infection states of the vertices over time—even this can be relaxed, as discussed above. While our theoretical guarantees require that the process has reached meta-stability, we show in Appendix A.2 that SISLearn achieves an F1 score of 0.80 with just 400 rounds of data, without needing to wait for meta-stability to be reached.
>
> ### References
>
> Gomez-Rodriguez M, Leskovec J, Krause A. Inferring Networks of Diffusion and Influence. *ACM Trans Knowl Discov Data*. 2012.
>
> Netrapalli P, Sanghavi S. Learning the graph of epidemic cascades. *SIGMETRICS Perform Eval Rev*. 2012.
>
> Tamaki H. Positive-instance driven dynamic programming for treewidth. *J Comb Optim*. 2019.

---

> > ### Comment · Reviewer_kFRe · 2025-04-03
> >
> > I am not very convinced about the response to the computational feasibility. You should acknowledge in the paper that the treewidth can be very large, so you really need the greedy algorithm also, in order to get polytime.
> >
> > For the stationarity part, it seems confusing because the paper starts with a discrete time process description, and then they make assumptions for this. That should be clarified

---

> > > ### Author Response · Authors · 2025-04-04
> > >
> > > ## Computational feasibility
> > > We thank the reviewer for their follow-up. We fully agree that the treewidth can be large in practice, and we will explicitly state in the revised manuscript that for graphs with $\text{tw}>20$, we recommend using the greedy algorithm. Our greedy method (Algorithm 4) is fast, scalable, and performs well on dense graphs with thousands of nodes, as demonstrated in new Figure 2.
> > >
> > > However, we believe it is important to emphasize two additional key insights regarding our DP algorithm, which further underscore the significance of our work:
> > >
> > > ### First polynomial-time algorithm for SRM on bounded-treewidth graphs:
> > >    Although the SRM problem is known to be NP-hard in general (Van Mieghem et al., 2011), we have shown—for the first time—that SRM can be solved optimally in polynomial-in-$n$ time on graphs with a bounded treewidth (lines 309–320). Specifically, if the treewidth is upper-bounded by **any** constant (independent of $n$), our DP algorithm runs in polynomial time in $n$. To the best of our knowledge, this was so far an open question. The solution to this question is a novel theoretical contribution of this paper, independent of the practical runtime of the DP algorithm.
> > >
> > > ### Significant practical speedup for exact SRM solutions:
> > >    Furthermore, even on graphs without constant treewidth bounds—where exponential runtime is inevitable for exact solutions—our DP algorithm is **five** orders of magnitude faster than the previously known approach for exactly solving SRM, as demonstrated in Figure 7 of our newly conducted simulations. Note that this practical runtime advantage **includes** the full end-to-end process: computing the treewidth, decomposing the graph into a nice tree decomposition, and running our DP algorithm.
> > >
> > > To summarize, here is how we envision our methods being applied in practice to solve the Vaccinating an Unknown Graph (VUG) problem:
> > >
> > > 1. Observations are collected and the underlying graph is inferred using SISLearn.
> > > 2. A fast, polynomial-time algorithm (such as Theorem 2 of Korhonen & Lokshtanov, 2023) is used to estimate an upper bound on the graph’s treewidth.
> > > 3. If the estimated treewidth bound is below $20$, our DP algorithm provides an optimal solution efficiently. Otherwise, our scalable greedy heuristic serves as an effective alternative.
> > >
> > > We will clearly emphasize these considerations in our revised manuscript to prevent any confusion regarding computational feasibility, and we hope this addresses the reviewer's valid concerns.
> > >
> > > ## Stationarity
> > > We thank the reviewer for their attention to the stationarity assumption. To clarify, the paper consistently models the SIS process as a discrete-time Markov chain throughout. If the confusion stems from our referencing of works on continuous-time processes, we note that discrete- and continuous-time SIS models exhibit similar qualitative behavior under comparable parameter regimes.
> > >
> > > Regarding stationarity: we cannot in general assume that any Markov chain has a non-trivial stationary distribution. In fact, in our case, the only true stationary distribution is the absorbing all-zero state. However, it is well-established (e.g., Cator & Van Mieghem, 2013) that in parameter regimes above the epidemic threshold, SIS dynamics will either go extinct quickly, or enter a _meta-stable_ regime—where the distribution over configurations remains approximately constant for a long period prior to extinction.
> > >
> > > Our Assumption 3.1 formalizes this idea: we assume that samples are drawn during the meta-stable phase, where the distribution is stable enough to allow reliable estimation. We will make this point more explicit in the revised manuscript.
> > >
> > > To support this further, we conducted experiments (new Figure 3 in file linked below) showing that for the China flu graph of the main paper, the process typically reaches this stable regime within 50–200 steps, depending on infection parameters. As expected, processes below the epidemic threshold (shaded region) do not reach meta-stability and go extinct quickly.
> > >
> > > Finally, we note that even without assuming meta-stability, our learning algorithm performs well in practice: in all experiments, data collection begins at time step 1 without enforcing stationarity, demonstrating robustness to deviations from this assumption.
> > >
> > > ### Thank you again for your careful review and thoughtful comments. We believe we've addressed your main concerns, clarified all technical points, and demonstrated both the soundness and significance of our contributions. We would greatly appreciate it if you could reconsider your evaluation.
> > >
> > > **Link to new figures: https://drive.proton.me/urls/5N1PC0M6ZW#oCweKFODmBNn**
> > >
> > > ## References
> > > Korhonen, T. and Lokshtanov, T. An Improved Parameterized Algorithm for Treewidth. In *STOC 2023*. 2023.
> > >
> > > Cator, E. and Van Mieghem, P. Susceptible-infected-susceptible epidemics on the complete graph and the star graph: Exact analysis. In *Phys Rev E*. 2013.

---

### Official Review · Reviewer_WHtL · 2025-03-13

**Overall Recommendation:** 3

**Summary:**

The paper tackles the critical challenge of minimizing disease extinction time in Susceptible-Infected-Susceptible (SIS) models under unknown contact networks. The authors propose a two-stage framework:

- Network Inference: A novel inclusion-exclusion learning algorithm with provable sample complexity bounds
- Vaccination Optimization: (i) Optimal dynamic programming for bounded tree width graphs, (ii) Efficient greedy heuristic for general graphs

Experimental validation on real-world influenza outbreak data demonstrates superior performance over baseline methods.

**Claims And Evidence:**

•	Theoretical Foundations:
- Formal proofs for structure learning guarantees (Theorem 3.1)
- Spectral radius-extinction time relationship analysis (Theorem 4.1)
- Time complexity analysis of vaccination algorithms (Theorem 4.2)

•	Empirical Validation:
- Consistent outperformance on Beijing influenza transmission networks
- Robustness tests with varying observation data sizes
- Statistical significance analysis through confidence intervals

**Essential References Not Discussed:**

While comprehensive, the paper could engage with Graph neural networks for epidemic modeling (e.g., GraphSAGE)

**Experimental Designs Or Analyses:**

Experimental Setup:
- Simulated SIS dynamics on Beijing influenza contact network (2009)
- Parameters calibrated to real-world transmission rates
- Simulation on vaccination strategy

Key Findings:
- Proposed method reduces infection prevalence compared to degree centrality vaccination
- Spectral radius reduction with only a low amount of node vaccination
- Maintains performance stability across observation data sizes

**Methods And Evaluation Criteria:**

•	Proposed Methodology:
- Structure Learning: Leverages infection state correlations via inclusion-exclusion principle
- Vaccination Strategies: (1) Dynamic programming using tree decomposition (bounded treewidth) (2) Greedy vertex removal guided by spectral impact (general graphs)

•	Evaluation Framework:
- Dataset: Augmented OutbreakTrees network with probabilistic edges
- Metrics: Infection proportion dynamics, spectral radius reduction
- Baselines: Comparative analysis against degree/random vaccination strategies

**Other Comments Or Suggestions:**

NA

**Other Strengths And Weaknesses:**

- Model Extensions: Incorporate SIR or SEIR dynamics and heterogeneous vaccination effects
- Algorithm Enhancement: Explore graph neural networks for large-scale networks
- Practical Considerations: Address implementation constraints (e.g., phased vaccination)

**Questions For Authors:**

- How does SIS Learn perform when infection vary over time? Could adaptively improve robustness to such changes?
- For graphs with tree width ω = 10 and size n = 1000, does the DP approach remain computationally feasible? What are the practical runtime limits?
- Given the success of graph neural networks (GNNs) in intervention strategies, why were they not included as baselines? Were there computational or methodological constraints?

**Relation To Broader Scientific Literature:**

This work innovatively bridges two research domains:
- Extends Ising model techniques to SIS dynamics
- Epidemic Control: Advances spectral radius minimization (SRM) approaches by removing prior structural knowledge requirements
- Distinct from existing SRM literature that assumes known networks, this study addresses realistic partial observation scenarios.

**Theoretical Claims:**

Key theoretical contributions:
- Theorem 3.1: Establishes polynomial sample complexity for graph recovery
- Theorem 4.1: Proves spectral radius directly impacts epidemic extinction time
- Theorem 4.2: Demonstrates time complexity in treewidth size
- Proofs employ established techniques from spectral graph theory and Markov processes, with detailed derivations in supplementary materials.

---

> ### Author Rebuttal · Authors · 2025-03-31
>
> We thank the reviewer for their positive assessment and their questions. We answer them below.
>
> ### Adaptivity
> Our learning algorithm, SISLearn, relies on data drawn from the meta-stable distribution of the SIS process, where the probabilities of configurations do not change anymore. However, it can accommodate time-varying infection probabilities, provided the changes are infrequent enough for the system to reach a new meta-stable state between shifts. As shown in Figure 3 of our additional experiments [here](https://drive.proton.me/urls/5N1PC0M6ZW#oCweKFODmBNn) (also see our response to Q6 of Reviewer Gdj4), the meta-stable state is reached in as few as 50 time steps depending on the SIS parameters.
>
> Importantly, since SISLearn sequentially learns each node’s neighborhood independently, even if the infection parameters shift mid-process, it suffices to re-estimate the new infection probabilities and allow the process to reach the new meta-stable state. Under these conditions, the theoretical guarantees would hold with minor changes. It should be noted that in practice, SISLearn performs well even without waiting for full stabilization. All experiments (main paper and appendix) use observations starting from the first round, and the algorithm still achieves high learning performance (see experiments in Appendix A.2).
>
> Moreover, SISLearn is robust to parameter misspecification. It only requires an estimate of the infection probability, and even if this input is significantly incorrect, the algorithm performs well in practice. In new experiments where the true infection probability was 0.3 but the input was set to 0.7, SISLearn still achieved an F1 score of 0.80 (compared to 0.97 with the correct value), showing graceful degradation under substantial misspecification. These new experiments are shown in Figure 4 [here](https://drive.proton.me/urls/5N1PC0M6ZW#oCweKFODmBNn).
>
> ### Computational feasibility
> We conducted two additional experiments to explicitly evaluate the computational feasibility of our DP algorithm. First, we measured runtime on graphs with fixed treewidth $\omega = 10$ and increasing number of vertices (up to $n=2000$); results provided in Figure 1 [here](https://drive.proton.me/urls/5N1PC0M6ZW#oCweKFODmBNn). These confirm that DP scales cubically in $n$, consistent with our theoretical analysis (Section D.3.3), and remains practical even for graphs with as many as $2000$ vertices!
>
> Second, we analyzed runtime on random Erdős–Rényi graphs with varying $n$ and naturally increasing treewidth; results provided in Figure 2 [here](https://drive.proton.me/urls/5N1PC0M6ZW#oCweKFODmBNn). This experiment highlights that while DP runtime grows exponentially with increasing treewidth, our proposed heuristic (Algorithm 4) remains computationally efficient (scaling cubically as shown in Section 4.3) and achieves vaccination performance close to DP, as demonstrated in Section 5.
>
> Thus, in practice, for graphs with known small-to-moderate treewidth ($\omega \lesssim 20$), the DP method is recommended even for large graphs. Otherwise, the greedy heuristic is an effective and scalable alternative, offering near-optimal performance with significantly lower runtime.
>
> ### Regarding GNNs
> While GNNs have shown success in epidemic modeling (Liu, 2024), to the best of our knowledge, no existing work directly employs GNNs for learning or vaccination strategies in SIS models. Developing such a GNN-based baseline would itself constitute a substantial research effort beyond a simple baseline comparison. However, if the reviewer is aware of specific GNN-based approaches applicable to our setting, we would gladly include them in our evaluation.
>
>
> **Link to new figures: https://drive.proton.me/urls/5N1PC0M6ZW#oCweKFODmBNn**
>
> ### References
> Liu Z, Wan G, Prakash BA, Lau MSY, Jin W. A Review of Graph Neural Networks in Epidemic Modeling. *Proceedings of the 30th ACM SIGKDD Conference on Knowledge Discovery and Data Mining.* KDD ’24.

---

### Official Review · Reviewer_Hjpv · 2025-03-17

**Overall Recommendation:** 2

**Summary:**

This paper studies the SIS model on an unknown graph. The process runs in discrete time. Any infected node becomes infected/susceptible in the next round with prescribed probabilities (independently of the rest of the graph). Any susceptible node can become infected from one of its infected neighbors. The goal is to vaccinate K individuals in a way to minimize the expected extinction time of the illness. When a vertex is vaccinated, the probability that it gets infected by a neighbor is decreased by a factor of $\alpha$ for all future rounds. Given a sequence of infection observations in discrete time, the authors propose an algorithm which first learns the graph and then decides which vertices to vaccinate. The graph learning step uses an inclusion-exclusion approach, first learning a superset of each vertex's neighbor set, and then paring it down using conditional independence properties of the infection process. The vaccination step solves a surrogate problem, namely Spectral Radius Minimization. The problem is solved in polynomial time for graphs with bounded treewidth. Experiments on a real dataset are included, comparing the proposed approach to several baselines.

**Claims And Evidence:**

Theorem 3.1 claims that under certain assumptions on the dataset (namely, the number of samples satisfying certain conditions, as well as a stationarity assumption), the inclusion-exclusion approach identifies the correct graph. Algorithm 2 provides a DP algorithm which runs in polynomial time (as long as the graph has bounded treewidth) and returns a set of K vertices whose deletion yields a graph with minimal spectral radius. The authors argue that solving this auxiliary problem is a good substitute for minimizing the expected extinction time.

**Essential References Not Discussed:**

N/A

**Experimental Designs Or Analyses:**

(See above concerns about the number of timesteps and the missing verification of graph estimation)

**Methods And Evaluation Criteria:**

The datasets and benchmarks are reasonable. Additional experiments are found in the supplementary material.
I found it hard to interpret the number of rounds in real-world terms. What would 2000 rounds correspond to in real time? If that is e.g. 100 years, then I don't find the stationarity assumption justified, since reducing the number of rounds significantly degrades the performance of the proposed algorithm (Figure 2).
Also, since the first step in the algorithm is estimating the graph, there should be an empirical verification of graph recovery.

**Other Comments Or Suggestions:**

N/A

**Other Strengths And Weaknesses:**

The paper is well-written.

**Questions For Authors:**

N/A

**Relation To Broader Scientific Literature:**

Epidemic modeling is a popular topic in network science. I am not aware of other results that require learning the network, so this is an interesting direction.

**Theoretical Claims:**

Lemma D.1 is false. Consider the case where 0 < p <= 1/2. Suppose that X_1 = 0 with probability 1/2 and X_1 = 2p with probability 1/2. Suppose all the X_i's are equal. Then $p = \frac{1}{n} \mathbb{E}[S_n]$. Set $\epsilon = p/2$ and $\delta < 1/2$. Then $\mathbb{P}\left(\left| \frac{1}{n} S_n - p \right| \geq \epsilon + \frac{1}{n} \right) = \frac{1}{2}$ for $n$ sufficiently large, contradicting the claim.
Theorem 1.8 of Pelekis and Ramon (2017; arXiv version) is false with the same counterexample. The journal version appears to be missing the corresponding result.

(Update below in light of rebuttal; score increased accordingly)

---

> ### Author Rebuttal · Authors · 2025-03-31
>
> We sincerely thank the reviewer for their careful reading and attention to detail. The reviewer is absolutely right—Lemma D.1, as originally stated, was incorrect and developed too hastily. We have corrected the result below by applying Theorem 1.2 from Kontorovich & Ramanan (2008). We also address the reviewer's other concerns below.
>
> ### Lemma D.1 (Replacement)
>
> Given observations $\\{Y^{(t)}\\}_{t \in T_D}$ from the SIS process where $T_D$ are not necessarily consecutive time indices and $Y^{(t)} \in\\{0,1\\}^V$, under Assumption 3.1, for any subset $U \subseteq V$, any state $y_U \in \\{0,1\\}^{|U|}$, and any positive $\varepsilon$ and $\delta$, we have the following deviation bound on the unbiased estimator
> $$
> \mathbb{P}\left(\left| \mathbb{P}(Y_U = y_U) - \hat{\mathbb{P}}(Y_U = y_U) \right| \geq \varepsilon \right) \leq\delta,
> $$
> whenever $|T_D| \geq \frac{2\log(2 / \delta)}{\varepsilon^2 (1-\theta)^2}$, where $0 < \theta < 1$ is a constant depending on the graph structure and model parameters.
> Here, $\mathbb{P}(Y_U = y_U)$ is the marginal probability of the vertices in $U$ being described by state vector $y_U$ and $\hat{\mathbb{P}}(Y_U = y_U) = \varphi_U$ is the empirical estimate over $T = |T_D|$ samples.
>
> Proof:
> Let $\mathcal{S}$ be the state space of our Markov chain, i.e., $\mathcal{S} = \\{0,1\\}^V$,  $T = |T_D|$ the number of observations, and $\\mathcal{S}^T$ be the product state space of $T$ observations. By definition, the process $\\{Y^{(t)}\\}\_{t \in \mathbb{N}} $ has the Markov property in time, hence so does a sub-sequence $\\{Y^{(t)}\\}\_{t \in T_D}$.
> We define $\varphi_U:\mathcal{S}^T \rightarrow \mathbb{R}$ as:
> $$
> \varphi_U(Y^1, \dots, Y^T) = \frac{1}{T} \sum_{t=1}^{T} \mathbb{1} \\{Y_U^{(t)} = y_U\\},
> $$
> where $ \mathbb{1} \\{\cdot\\}$ is the indicator function.
>
> Notice that this function is $1/T$-Lipschitz with respect to the Hamming metric $d(X,Y) = \sum_{t=1}^T \mathbb{1} \\{X^{(t)} \neq Y^{(t)}\\}$ on $\mathcal{S}^T$.
>
> We apply Thm. 1.2 by Kontorovich & Ramanan (2008) for Markov chains, stating that for a $c$-Lipschitz function $\varphi$ on $\mathcal{S}^n$,
> $$
> \mathbb{P}\\{|\varphi-\mathbb{E} \varphi| \geq t\\} \leq 2 \exp \left(-\frac{t^2}{2 n c^2 M_n^2}\right).
> $$
> In our case, the sequence length is $T$ (replacing $n$ in the theorem), $c=1/T$, $M_n$ becomes $M_T$, and $t$ becomes $\varepsilon$. The bound then becomes
> $$
> \mathbb{P}\\{|\varphi_U-\mathbb{E} \varphi_U| \geq \varepsilon\\} \leq 2 \exp \left(-\frac{\varepsilon^2}{2 T (1/T)^2 M_T^2}\right) = 2 \exp \left(-\frac{T \varepsilon^2}{2 M_T^2}\right),
> $$
> where $M_T = (1-\theta^T)/(1- \theta)$ and $\theta$ is the Markov contraction coefficient given by
> $$
> \theta = \sup\_{\\{Y^{\prime}, Y^{\prime \prime} \in \mathcal{S} \\}} \left\\|p\left(\cdot \mid Y^{\prime}\right)-p\left(\cdot \mid Y^{\prime \prime}\right)\right\\|_{\mathrm{TV}}.
> $$
> We have $\theta <1$ since every configuration $Y$ can transfer into the all-zero configuration $\underline{0}$. Hence, for any two states $Y^{\prime}, Y^{\prime \prime} \in \mathcal{S}$ both $p(\underline{0}|Y^{\prime})>0$ and $p(\underline{0}|Y^{\prime \prime})>0$ and the support of $p(\cdot|Y^{\prime})$ and $p(\cdot|Y^{\prime \prime})$ is not disjoint which implies $\theta <1$.
>
> Therefore, $M_T <1/(1-\theta)$ and the probability bound becomes:
> $$
> \mathbb{P}\\{|\varphi_U-\mathbb{E} \varphi_U| \geq \varepsilon\\} < 2 \exp\left(-\frac{T \varepsilon^2 (1-\theta)^2}{2}\right).
> $$
>
> Finally, setting the RHS to be $\leq \delta$ and solving for $T$ we get
> $$
> T \geq \frac{2\log(2 / \delta)}{\varepsilon^2 (1-\theta)^2}. \quad \square
> $$
>
> We emphasize that Lemma D.1 is not a central result of our paper. It is used solely to derive concentration bounds for estimating the direct and conditional influence quantities, thereby enabling our sample complexity analysis. Crucially, the correctness of the SISLearn algorithm itself is unaffected by this lemma. With the corrected bound, all downstream results remain valid (modulo minor constant adjustments), and in fact, Lemmas D.2–D.4 become cleaner, as we can eliminate the $1/m$ term in favor of a dependence on the (constant) Markov contraction coefficient $\theta$, which depends on the graph structure and the infection parameters.
>
> ### Regarding rounds
> The real-world interpretation of a "round" depends on the application. It may correspond to seconds in financial or communication networks, or days in epidemiological settings. The time to reach stationarity similarly depends on the timescale of the underlying process.
>
> ### Regarding graph recovery
>
> We provide experimental results on the learning performance of SISLearn in App. A.2. We also performed new experiments on the robustness of SISLearn, given in new Fig. 4 [here](https://drive.proton.me/urls/5N1PC0M6ZW#oCweKFODmBNn) (see Q4 of Reviewer Gdj4).
>
> ### References
>
> Kontorovich L., Ramanan K. Concentration inequalities for dependent random variables via the martingale method. *The Annals of Probability*. 2008.

---

> > ### Comment · Reviewer_Hjpv · 2025-04-05
> >
> > (copying as it was originally posted as an "official comment")
> > Thank you for addressing my main concern regarding Lemma D.1. An observation: your upper bound on \theta is implicitly 1 minus the probability of reaching the all zero state, from a worst-case initial state. Then 1-\theta is upper-bounded by the worst-case probability of reaching the all-zero state =: q. The extinction time is dominated by Geom(q), which means that the theoretical guarantee for T is larger than the square of the expected extinction time. This means that the strategy of learning the graph first does not come with a meaningful theoretical guarantee (as extinction would likely happen before T steps), though it might work well in practice.

---

> > > ### Author Response · Authors · 2025-04-07
> > >
> > > We thank the reviewer for their follow-up and observation connecting our sample complexity bound to extinction time.
> > >
> > > As the reviewer points out, in the worst case, $\theta = 1 - q$. However, this worst-case $\theta$ is governed by transitions from extinction (i.e., the all-zero state $\underline{0}$) and does **not** reflect the behavior of the chain in the meta-stable regime. Since the problem becomes trivial upon extinction, we are implicitly conditioning on survival. If one restricts the supremum in the definition of $\theta$ to configurations in $\\{0,1\\}^V \setminus \\{\underline{0}\\}$, the resulting contraction coefficient becomes strictly smaller, depending instead on the graph structure and infection parameters.
> > >
> > > Turning to our bound on $T$, we wish to clarify that the direction of the inequality cited by the reviewer is reversed. Our guarantee requires $T \geq \frac{C}{(1 - \theta)^2}$, where $C = \frac{2 \log(2 / \delta)}{\varepsilon^2}$. Since $\theta <1 - q$, it follows that $(1 - \theta)^2 > q^2$, and thus $\frac{1}{(1 - \theta)^2} < \frac{1}{q^2}$. Therefore, our required number of samples $T$ is **upper bounded** by $\mathcal{O}(1/q^2)$—not larger than it. In other words, contrary to the reviewer’s conclusion, our theoretical sample complexity is at most on the order of the square of the expected extinction time, not worse.
> > >
> > > These observations clarify that our theoretical bound is meaningful within the intended regime—i.e., while the process has not gone extinct and remains in its meta-stable phase. In this regime, the relevant contraction coefficient is smaller than the worst-case bound. Indeed, our experiments confirm that SISLearn performs well even with short observation windows (e.g., 400 rounds starting from $t = 1$, without waiting for meta-stability), supporting the practical relevance of the bound.
> > >
> > > We hope this clarification resolves the concern, and we sincerely thank the reviewer again for their thoughtful engagement with the technical details.

---

### Official Review · Reviewer_Gdj4 · 2025-03-24

**Overall Recommendation:** 3

**Summary:**

This paper is broadly about vaccinating the nodes of a network over time (subject to a total budget on the number of vaccinations) in order to minimize the expected extinction time of the epidemic. The SIS model is assumed, and interestingly, the network is not assumed to be known, but has to be learned. (SIS is a reasonable model where individuals can get infected repeatedly, such as with cholera.) The paper thus splits the task into two parts: (a) learning the network, and (b) vaccination. Theoretical results and experimental evidence are given.

## update after rebuttal: I have increased my score to "Weak Accept" given the rebuttals.

**Claims And Evidence:**

I find the following claims problematic/unclear: I ask the authors to clearly explain these.

1. In what sense is the algorithm "optimal" as claimed in the abstract? If it is in the sense of "if the underlying graph is a tree" as in Appendix B, this would be a very weak claim as trees are not at all natural models for disease/communication spread.

2. I assume the algorithm is not "online" in the sense that for each t, given the infection states up to time t, we have to immediately decide the set R_t of nodes to vaccinate at time t? That is, we assume the "offline" case where all the Y^t (for 1 <= t <= T) are given upfront? If so, why do the vaccination in multiple stages at all?

3. It does not seem a reasonable assumption to make that all nodes independently get infected with the same probability at time t = 0: how flexible can this be made?

4. I assume the disease parameters are known to the algorithm?

5. Give references in the literature to where social networks seem to have small treewidth.

6. Please justify the stationarity assumption: it appears strong to me.

**Essential References Not Discussed:**

N/A to my knowledge.

**Experimental Designs Or Analyses:**

The experimental analysis appears adequate.

**Methods And Evaluation Criteria:**

The methods appear reasonable.

**Other Comments Or Suggestions:**

Say *expected* in the initial discussion on extinction time as well, as you have done in defining the VUG problem.

The paper is well-written in general. One typo: "combines network learning with strategic vaccination strategy" -->  "combines network learning with a strategic vaccination strategy"

**Other Strengths And Weaknesses:**

Learning the network appears like a good problem to me. The vaccination model parametrized by alpha also looks good and reasonable.

**Questions For Authors:**

Please respond in detail to my modeling/other questions.

**Relation To Broader Scientific Literature:**

The connections to the existing literature look adequate to me.

**Theoretical Claims:**

I have asked the authors to justify the stationarity assumption.

---

> ### Author Rebuttal · Authors · 2025-03-31
>
> We thank the reviewer for their detailed review and their insightful questions. We respond to each of them in detail below.
>
> ### Q1
> We propose three vaccination strategies, all of which solve the Spectral Radius Minimization (SRM) problem:
>
> (1) An optimal polynomial-time algorithm for trees (Appendix B, Algorithm 5).
>
> (2) An optimal (DP) algorithm that solves SRM on arbitrary graphs (Section 4.2, Algorithm 2), with polynomial runtime on graphs of bounded treewidth.
>
> (3) A greedy polynomial-time heuristic (Section 4.3, Algorithm 4).
>
> The abstract refers to the second method. By “optimal,” we mean that it exactly solves the SRM problem: given a budget $K$, it finds a subset of vertices of size $\leq K$ whose removal minimizes the spectral radius. This is stated and proven in Theorem D.1.
>
> ### Q2
> The VUG problem is formulated as an online problem: the agent observes the infection states over time and must decide when and whom to vaccinate, subject to a global budget $K$ (Section 2). In particular, the agent is not required to vaccinate all $K$ nodes at once.
>
> That said, our proposed method instantiates this framework in an offline fashion: we collect $T_D$ rounds of observations, learn the graph using SISLearn, and then vaccinate all $K$ nodes at time $t = T_D + 1$. This choice reflects that, assuming the graph is learned perfectly, vaccinating all at once is optimal for minimizing extinction time.
>
> Exploring *adaptive* or *staggered* vaccination—where the agent begins intervening while still learning—is a compelling direction for future work.
>
> ### Q3
> This assumption is not essential for our approach or theoretical results. All our lemmas, theorems, and algorithmic results hold under different initial infection states, whether they involve a single infected vertex, an arbitrary deterministic or probabilistic subset, or any other initial configuration. We adopted the standard uniform-seed initialization solely because it is common in the SIS literature.
>
> ### Q4
> The only disease parameter SISLearn requires is the infection probability, $p_\text{inf}$, which can be estimated from observational data, like done in Kirkeby et al. (2017) via a mean-field approximation. Importantly, SISLearn is robust to parameter misspecification, as evident by new experiments where the true infection probability was 0.3 but the input was set to 0.7, and SISLearn still achieved an F1 score of 0.80 (compared to 0.97 with the correct value). These new experiments are shown in new Figure 4 (see linked file below).
>
> ### Q5
> While real-world social networks may not exhibit small treewidth, our work addresses this directly: our DP algorithm is ideal for graphs with small to moderate treewidth (e.g., up to ~10-20; see new Figure 1, and our response to Reviewer WHtL), while our fast and scalable greedy heuristic (Algorithm 4) performs well on general graphs—including those with high density (see new Figure 2).
>
> ### Q6
> Our assumption of stationarity arises naturally from well-established theoretical results for SIS-type processes. It has been shown that if the infection parameters are above the epidemic threshold $\rho(\mathcal{G}) \geq p_{\text{rec}}/p_{\text{inf}}$, the process may reach a meta-stable distribution, resembling a stationary distribution (see, among others, Schonmann 1985, Liggett 1999, and Mountford et al. 2013). Conversely, if the threshold condition is not satisfied, the infection dies out rapidly, trivially resolving the vaccination problem.
>
> More formally, following the coupling argument of Cator and Van Mieghem (2013), the SIS process can be related to a modified Markov chain that excludes the absorbing all-zero state. This modified chain is ergodic and therefore has a proper stationary distribution. Thus, until extinction occurs, the original SIS process behaves like an ergodic Markov chain with a stationary distribution.
>
> In other words, either the SIS process (quickly) reaches a meta-stable state where our assumption holds, or it dies out. The former is backed up by new experiments, given in new Figure 3, where the SIS process reached the meta-stable state in an average of 100 rounds.
>
> Thus, the stationarity assumption, while seemingly strong, is both theoretically well-founded and empirically justified.
>
> ### New figures: https://drive.proton.me/urls/5N1PC0M6ZW#oCweKFODmBNn
> ### References
>
> Kirkeby C, et al. Methods for estimating disease transmission rates: Evaluating the precision of Poisson regression and two novel methods. *Sci Rep*. 2017.
>
> Cator E, et al. Susceptible-infected-susceptible epidemics on the complete graph and the star graph: Exact analysis. *Phys Rev E*. 2013.
>
> Liggett TM. Stochastic Interacting Systems: Contact, Voter and Exclusion Processes. *Springer*. 1999.
>
> Mountford T, et al. Metastable densities for the contact process on power law random graphs. *Electronic Journal of Probability*. 2013.
>
> Schonmann RH. Metastability for the contact process. *J Stat Phys*. 1985.

---

### Decision · Program_Chairs · 2025-05-01

**Decision:**

Accept (poster)

**Comment:**

This paper is broadly about vaccinating the nodes of a network over time (subject to a total budget on the number of vaccinations) in order to minimize the expected extinction time of the epidemic. The Susceptible-Infected-Susceptible (SIS) model is assumed, and the network is not assumed to be known, but has to be learned. (SIS is a reasonable model where individuals can get infected repeatedly, such as with cholera.) The paper thus splits the task into two parts: (a) learning the network, and (b) vaccination. Theoretical results and experimental evidence are given.

Inferring the network is a nice and practical problem.

The Spectral Radius Minimization (SRM) problem is NP-hard in general: the paper shows that it can be solved in polynomial time on networks with a bounded treewidth (the paper also suggests an alternative greedy heuristic in practice when the treewidth exceeds 20, say). While this is a reasonable theoretical contribution, several NP-hard graph problems become tractable when the treewidth is held constant---and hence I would not rate this as a primary major contribution of the paper.

The authors say in a rebuttal “Either the process dies out quickly, in which case vaccination is unnecessary, or it enters a meta-stable regime where it behaves like a Markov chain with a stationary distribution, justifying Assumption 3.1.” While this seems reasonable, a meta-stability assumption seems very strong: e.g., what if the time to meta-stability is very long (e.g., if the dynamics don’t have a reasonable mixing time)? Parametrizing perhaps by time to reach a meta-stable regime will make the setting more realistic.